# Provably Efficient Offline Reinforcement Learning in Regular Decision Processes

**Roberto Cipollone**
Sapienza University of Rome
cipollone@diag.uniroma1.it

**Anders Jonsson**
Universitat Pompeu Fabra
anders.jonsson@upf.edu

**Alessandro Ronca**
University of Oxford
alessandro.ronca@cs.ox.ac.uk

**Mohammad Sadegh Talebi**
University of Copenhagen
m.shahi@di.ku.dk

## Abstract

This paper deals with offline (or batch) Reinforcement Learning (RL) in episodic Regular Decision Processes (RDPs). RDPs are the subclass of Non-Markov Decision Processes where the dependency on the history of past events can be captured by a finite-state automaton. We consider a setting where the automaton that underlies the RDP is unknown, and a learner strives to learn a near-optimal policy using pre-collected data, in the form of non-Markov sequences of observations, without further exploration. We present `RegORL`, an algorithm that suitably combines automata learning techniques and state-of-the-art algorithms for offline RL in MDPs. `RegORL` has a modular design allowing one to use any off-the-shelf offline RL algorithm in MDPs. We report a non-asymptotic high-probability sample complexity bound for `RegORL` to yield an $\varepsilon$-optimal policy, which makes appear a notion of concentrability relevant for RDPs. Furthermore, we present a sample complexity lower bound for offline RL in RDPs. To our best knowledge, this is the first work presenting a provably efficient algorithm for offline learning in RDPs.

## 1 Introduction

Most reinforcement learning (RL) algorithms hinge on the Markovian assumption, i.e. that the underlying system transitions and rewards are Markovian in some natural notion of (observable) state, and hence, the distribution of future observations depends only on the current state-action of the system. This fundamental assumption allows one to model decision making using the powerful framework of Markov Decision Processes (MDPs) [1]. However, there are many application scenarios where rewards are issued according to temporal conditions over histories (or trajectories), and others where the environment itself evolves in a history-dependent manner. As a result, Markovian approaches may prove unsuitable for modeling such situations. These scenarios can be appropriately modeled as *Non-Markovian Decision Processes (NMDPs)* [2, 3].

NMDPs describe environments where the distribution on the next observation and reward is a function of the history. In these environments, behaving optimally may also require to take histories into account. For example, a robot may receive a reward for delivering an item only if the item was previously requested, and a self-driving car is more likely to skid and lose control if it previously rained. Also, consider a mobile robot that has to track an object which may disappear from its field of view. The object is likely to be found again in the same place where it was seen last time. This requires the agent to remember, hence to act according to information in its interaction history. In general, an NMDP can show an arbitrary dependency on the history or trace, preventing efficient learning. Consequently, recent research has focused on tractable sub-classes of NMDPs. In Regular

Decision Processes (RDPs) [3], the next observation and reward distributions depend on regular properties of the history, which can be captured by a deterministic finite-state automaton. This determines the existence of a finite state space where states are determined by histories, and where the Markov property is regained.

In this paper, we investigate offline RL in episodic RDPs, where the goal is to find a near-optimal policy using a pre-collected dataset, with minimal possible size, generated by a fixed behavior policy (and without further exploration). Offline RL in MDPs has received extensive attention recently, and provably sample efficient algorithms have been proposed for various settings. Despite the extensive and rich literature on MDPs, comparatively little work exists on offline RL in NMDPs. The scarcity of results may likely be attributed to the difficult nature of the problem rather than the lack of interest.

Partially-Observable Markov Decision Processes (POMDPs) [4] are also NMDPs, and RDPs can be seen as the subclass of POMDPs that enjoy the property of having hidden states determined by the history of observations. This is a key property that allows one to take advantage of a set of planning and learning techniques that do not apply to arbitrary POMDPs. Planning in POMDPs is computationally intractable [5], and two common approaches to solve (and learn) them rely on maintaining either a belief state or a finite history of observations. Maintaining and updating a belief state is worst-case exponential in the size of the original observation space, while the latter approach yields a space whose size is exponential in the history length. State-of-the-art work on offline RL in POMDPs considers restricted classes of POMDPs such as undercomplete POMDPs (e.g., [6, 7]), which cannot be used to model all RDP instances. General POMDPs are only considered under assumptions such as the possibility of reaching every belief state in a few steps [8] or ergodicity [9]. While existing offline RL algorithms for solving POMDPs cannot guarantee provable learning in a generic RDP, the structural properties of RDPs indicate that they can be solved more efficiently using techniques that are carefully tailored to their structure. Exploiting the structure in RDPs is thus key in designing provably sample-efficient learning algorithms.

## 1.1 Summary of Contributions

We formalize offline RL in RDPs (Section 2), and establish a first, to the best of our knowledge, sample complexity lower bound thereof (Section 5). We introduce an algorithm, called `RegORL`, that learns $\varepsilon$-optimal policies for any RDP, in the episodic setting. At the core of `RegORL`, there is a component called ADACT–H, which is a variant of ADACT [10], carefully tailored to episodic RDPs. ADACT–H learns a minimal automaton that underlies the unknown RDP without prior knowledge. The output automaton is further used to derive a Markov abstraction of data to be used by any off-the-shelf algorithm for offline RL in episodic MDPs. We present a sample-complexity bound for ADACT–H to return a minimal underlying automaton with high probability. This bound substantially improves the existing bound for the original ADACT, and can be of independent interest. In view of the modular design of `RegORL`, the total sample complexity is controlled by twice that of ADACT–H (Theorem 6) and that for the incorporated off-the-shelf algorithm. We also present another variant of ADACT–H, called ADACT–H–A. In contrast to ADACT–H that learns a complete RDP, ADACT–H–A only reconstructs a subset of states that are likely under the behavior policy, in relation to an input accuracy parameter. As such, ADACT–H–A renders more aligned with the practice of RL than ADACT–H. Furthermore, we provide a first lower-bound for offline RL in RDPs that involves relevant parameters for the problem, such as the RDP single-policy concentrability, which extends an analogous notion for MDPs from the literature. Finally, if contrasted to both online learning in RDPs and automata learning, our results suggest possible improvements in sample complexity results for both areas.

## 1.2 Related Work

**Offline RL in MDPs.** There is a rich and growing literature on offline RL, and provably sample efficient algorithms have been proposed for various settings of MDPs; see, e.g., [11, 12, 13, 14, 15, 16, 17, 18, 19, 20]. For example, in the case of episodic MDPs, it is established that the optimal sample size in offline RL depends on the size of state-space, episode length, as well as some notion of concentrability, reflecting the distribution mismatch between the behavior and optimal policies. A closely related problem is off-policy learning; see, e.g., [21, 22, 23] and the recent survey [24].

**Online RL in RDPs.** Several algorithms for *online* RL in RDPs exist [25, 26, 27] but complexity bounds are only given in [26] for the infinite-horizon discounted setting. The sample complexity bounds in [26] are not immediately comparable to ours, due to the different setting. Importantly, the algorithm in [26] uses the uniform policy for learning, and it therefore might be adapted to our setting only under the assumption that the behaviour policy is uniform. Even in this case, our bounds show an improved dependency on several key quantities. Furthermore, we provide a sample complexity lower bound, whereas their results are limited to showing that a dependency on the quantities occurring in their upper bounds is necessary.

The online RL algorithms in [28, 29, 30, 31] have been developed for formalisms that are closely related to RDPs, and such algorithms can be applied to RDPs. However, these algorithms are not proven to be sample efficient.

**POMDPs.** Every RDP can be seen as a POMDP whose hidden dynamics evolves according to its finite-state automaton. However, RL in POMDPs is a largely open problem. Even for a known POMDP, computing a near-optimal policy is PSPACE-complete [5]. For unknown dynamics, which is the setting considered here, favourable bounds have been obtained for the class of undercomplete POMDPs [6, 7], which does not include all RDPs, or alternatively, under other assumptions such as few-step reachability [8] or ergodicity [9]. This relationship between RDPs and POMDPs can be also seen from the notion of state. In fact, the automaton state of an RDP is an instance of information state, as defined in [32], and of belief, as in classic POMDP literature [33].

**PSRs.** Predictive State Representations (PSRs) [34, 35, 36, 37] are general descriptions of dynamical systems that capture POMDPs and hence RDPs. There exist polynomial PAC bounds for online RL in PSRs [38]. Nonetheless, these bounds are looser than the one we show here, since they must necessarily consider a wider class of models. Moreover, although a minimum core set for PSRs is similar to a minimal RDP, the bounds feature a number of quantities that are specific to PSRs (e.g., regularity parameter) and do not immediately apply to RDPs.

**Other Non-Markovian Settings.** *Feature MDPs* and *state representations* both share the idea of having a map from histories to a state space. This is analogous to the map determined by the transition function of the automaton underlying an RDP. Algorithmic solutions for feature MDPs are based on suffix trees, and they cannot yield optimal performance in our setting [29, 30]. The automaton of an RDP can be seen as providing one kind of state representation [39, 40, 41, 42]. The existing bounds for state representations show a linear dependency on the number of candidate representations, which is exponential in the number of states in our case. A similar dependency is also observed in [43]. RL with *non-Markovian rewards* is considered in [44, 45, 46, 47, 48, 49]. The idea of a map from histories to states is also found in [32]. Non-Markovianity is also introduced by logical specifications that the agent is required to satisfy [50, 51, 52, 53, 54]; however, it is resolved a priori from the known specification. The convergence properties of Q-learning over a (known) underlying state space such as the one of an RDP are studied in [55].

**Learning PDFA.** Our algorithms for learning an RDP borrow and improve over techniques for learning Probabilistic-Deterministic Finite Automata (PDFA) [56, 57, 58, 10, 59, 60]. Our algorithm builds upon the state-of-the-art algorithm ADACT [10], and we derive bounds that are a substantial improvement over the ones that would be obtained from a straightforward application of any existing PDFA-learning algorithm to the offline RL setting.

We provide additional literature review in Appendix A.

## 2 Preliminaries and Problem Formulation

**Notations.** Given a set $\mathcal{Y}$, $\Delta(\mathcal{Y})$ denotes the set of probability distributions over $\mathcal{Y}$. For a function $f : \mathcal{X} \to \Delta(\mathcal{Y})$, $f(x, y)$ is the probability of $y$ given $x$. Further, we write $y \sim f(x)$ to abbreviate $y \sim f(x, \cdot)$. For $y \in \mathcal{Y}$, we use $\mathbb{1}_y \in \Delta(\mathcal{Y})$ to denote the Kronecker delta defined as $\mathbb{1}_y(y) = 1$ and $\mathbb{1}_y(y') = 0$ for each $y' \in \mathcal{Y}$ such that $y' \neq y$. Given an event $E$, $\mathbb{I}(E)$ denotes the indicator function of $E$, which equals 1 if $E$ is true, and 0 otherwise, e.g. $\mathbb{1}_y(y') = \mathbb{I}(y = y')$. For any integer $Z \geq 0$, we let $[Z] := \{0, \ldots, Z\}$. Given a set $\mathcal{X}$, for $k \in \mathbb{N}$, $\mathcal{X}^k$ represents the set of sequences of length $k$ whose elements are from $\mathcal{X}$. Also, $\mathcal{X}^* = \cup_{k=0}^{\infty} \mathcal{X}^k$. The notation $\widetilde{\mathcal{O}}(\cdot)$ hides poly-logarithmic terms.

## 2.1 Episodic Regular Decision Processes

We first introduce generic episodic decision processes. An episodic decision process is a tuple $\mathcal{P} = \langle \mathcal{O}, \mathcal{A}, \mathcal{R}, \bar{T}, \bar{R}, H \rangle$, where $\mathcal{O}$ is a finite set of observations, $\mathcal{A}$ is a finite set of actions, $\mathcal{R} \subset [0,1]$ is a finite set of rewards and $H \geq 1$ is a finite horizon. As is common in automata theory, we use sequences $a_m r_m o_m \cdots a_n r_n o_n$ to denote traces of actions, rewards and observations, and concatenation $\mathcal{A}\mathcal{R}\mathcal{O} = \{aro : a \in \mathcal{A}, r \in \mathcal{R}, o \in \mathcal{O}\}$ to denote sets of sequences. Let $\mathcal{E}_t = (\mathcal{A}\mathcal{R}\mathcal{O})^{t+1}$ be the set of traces of length $t+1$, and let $e_{m:n} \in \mathcal{E}_{n-m}$ denote a trace from time $m$ to time $n$, included. A *trajectory* $e_{0:T}$ is the full trace generated until time $T$. We assume that a trajectory $e_{0:T}$ can be partitioned into *episodes* $e_{\ell:\ell+H} \in \mathcal{E}_H$ of length $H+1$, and that the dynamics at time $T = k(H+1) + t, t \in [H]$, are *conditionally independent* of the previous episodes and all rewards, i.e. the dynamics only depend on $a_{k(H+1)} o_{k(H+1)} \cdots a_T o_T$. For $t \in [H]$, let $\mathcal{H}_t = (\mathcal{A}\mathcal{O})^{t+1}$ denote the relevant part of the trajectory for decision making, and let $\mathcal{H} = \cup_{t=0}^{H} \mathcal{H}_t$. We refer to elements in $\mathcal{H}$ as *histories*, even though they are not complete trajectories. In each episode $e_{0:H}$, $a_0 = a_\perp$ is a dummy action used to initialize the distribution on $\mathcal{H}_0$. The transition function $\bar{T} : \mathcal{H} \times \mathcal{A} \to \Delta(\mathcal{O})$ and the reward function $\bar{R} : \mathcal{H} \times \mathcal{A} \to \Delta(\mathcal{R})$ only depend on the history of the current episode. Given $\mathcal{P}$, a generic policy is a function $\pi : (\mathcal{A}\mathcal{O})^* \to \Delta(\mathcal{A})$ that maps trajectories to distributions over actions. The value function $V^\pi : [H] \times \mathcal{H} \to \mathbb{R}$ of a policy $\pi$ is a mapping that assigns real values to histories. For $h \in \mathcal{H}$, it is defined as $V^\pi(H, h) := 0$ and

$$V^\pi(t, h) := \mathbb{E}\left[ \sum_{i=t+1}^{H} r_i \,\middle|\, h, \pi \right], \quad \forall t < H, \forall h \in \mathcal{H}_t. \tag{1}$$

For brevity, we write $V_t^\pi(h) := V^\pi(t, h)$. The optimal value function $V^*$ is defined as $V_t^*(h) := \sup_\pi V_t^\pi(h), \forall t \in [H], \forall h \in \mathcal{H}_t$, where $\sup$ is taken over all policies $\pi : (\mathcal{A}\mathcal{O})^* \to \Delta(\mathcal{A})$. Any policy achieving $V^*$ is called optimal, which we denote by $\pi^*$; namely $V^{\pi^*} = V^*$. Solving $\mathcal{P}$ amounts to finding $\pi^*$. In what follows we consider simpler policies of the form $\pi : \mathcal{H} \to \Delta(\mathcal{A})$ mapping histories to distributions over actions. Let $\Pi_\mathcal{H}$ denote the set of such policies. It can be shown that $\Pi_\mathcal{H}$ always contains an optimal policy, i.e. $V_t^*(h) := \max_{\pi \in \Pi_\mathcal{H}} V_t^\pi(h), \forall t \in [H], \forall h \in \mathcal{H}_t$. An episodic MDP is an episodic decision process whose dynamics at each timestep $t$ only depends on the last observation and action [1].

**Episodic RDPs.**  An episodic Regular Decision Process (RDP) [3, 25] is an episodic decision process $\mathbf{R} = \langle \mathcal{O}, \mathcal{A}, \mathcal{R}, \bar{T}, \bar{R}, H \rangle$ described by a *finite transducer* (Moore machine) $\langle \mathcal{Q}, \Sigma, \Omega, \tau, \theta, q_0 \rangle$, where $\mathcal{Q}$ is a finite set of states, $\Sigma = \mathcal{A}\mathcal{O}$ is a finite input alphabet composed of actions and observations, $\Omega$ is a finite output alphabet, $\tau : \mathcal{Q} \times \Sigma \to \mathcal{Q}$ is a transition function, $\theta : \mathcal{Q} \to \Omega$ is an output function, and $q_0 \in \mathcal{Q}$ is a fixed initial state [61, 62, 63]. The output space $\Omega = \Omega_\mathsf{o} \times \Omega_\mathsf{r}$ consists of a finite set of functions that compute the conditional probabilities of observations and rewards, meaning $\Omega_\mathsf{o} \subset \mathcal{A} \to \Delta(\mathcal{O})$ and $\Omega_\mathsf{r} \subset \mathcal{A} \to \Delta(\mathcal{R})$. For simplicity, we use two output functions, $\theta_\mathsf{o} : \mathcal{Q} \times \mathcal{A} \to \Delta(\mathcal{O})$ and $\theta_\mathsf{r} : \mathcal{Q} \times \mathcal{A} \to \Delta(\mathcal{R})$, to denote the individual conditional probabilities. Let $\tau^{-1}$ denote the inverse of $\tau$, i.e. $\tau^{-1}(q) \subseteq \mathcal{Q} \times \mathcal{A}\mathcal{O}$ is the subset of state-symbol pairs that map to $q \in \mathcal{Q}$. In this context, an input symbol is an element of $\mathcal{A}\mathcal{O}$. We use $A, R, O, Q$ to denote the cardinality of $\mathcal{A}, \mathcal{R}, \mathcal{O}, \mathcal{Q}$, respectively, and assume $A \geq 2$.

An RDP $\mathbf{R}$ implicitly represents a function $\bar{\tau} : \mathcal{H} \to \mathcal{Q}$ from histories in $\mathcal{H}$ to states in $\mathcal{Q}$, recursively defined as $\bar{\tau}(h_0) := \tau(q_0, a_0 o_0)$ and $\bar{\tau}(h_t) := \tau(\bar{\tau}(h_{t-1}), a_t o_t)$. The dynamics and of $\mathbf{R}$ are defined as $\bar{T}(h, a, o) = \theta_\mathsf{o}(\bar{\tau}(h), a, o)$ and $\bar{R}(h, a, r) = \theta_\mathsf{r}(\bar{\tau}(h), a, r), \forall h \in \mathcal{H}, \forall aro \in \mathcal{A}\mathcal{R}\mathcal{O}$. Episodic RDPs are acyclic, i.e. the states can be partitioned as $\mathcal{Q} = \mathcal{Q}_0 \cup \cdots \cup \mathcal{Q}_{H+1}$, where each $\mathcal{Q}_{t+1}$ is the set of states generated by histories in $\mathcal{H}_t$ for each $t \in [H]$. An RDP is minimal if its Moore machine is minimal. Since there is nothing to predict at time $H+1$, a minimal RDP contains a single state $q_{H+1}$ in $\mathcal{Q}_{H+1}$. To ensure that an acyclic RDP $\mathbf{R}$ is minimal, we introduce a designated termination observation $o_\perp$ in $\mathcal{O}$ and define $\tau(q_{H+1}, ao_\perp) = q_{H+1}$ and $\theta_\mathsf{o}(q_{H+1}, a) = \mathbb{1}_{o_\perp}$ for each $a \in \mathcal{A}$. Hence, $q_{H+1}$ is absorbing and the states in $\mathcal{Q}$ implicitly count how many steps are left until we observe $o_\perp$. Without $o_\perp$, a Moore machine could potentially represent all episodes using fewer than $H+2$ states.

Since the conditional probabilities of observations and rewards are fully determined by the current state-action pair $(q, a)$, an RDP $\mathbf{R}$ adheres to the Markov property over its states, but *not over the observations*. Given a state $q_t \in \mathcal{Q}$ and an action $a_t \in \mathcal{A}$, the probability of the next transition is

$$\mathbb{P}(r_t, o_t, q_{t+1} \mid q_t, a_t, \mathbf{R}) = \theta_\mathsf{r}(q_t, a_t, r_t)\, \theta_\mathsf{o}(q_t, a_t, o_t)\, \mathbb{I}(q_{t+1} = \tau(q_t, a_t o_t)).$$

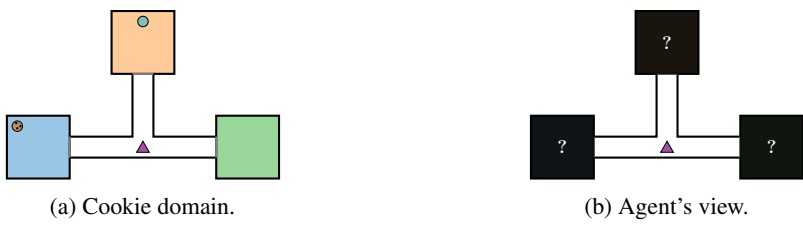



(a) Cookie domain.          (b) Agent's view.

Figure 1: The *cookie* domain: The agent can only see what is in the current room [28].



Evidently, in the special case where an RDP is Markovian in both observations and rewards, it reduces to an episodic MDP. More precisely, any episodic MDP with actions $\mathcal{A}$, states $\mathcal{O}$ and horizon $H$ can be represented by some episodic RDP with states $\mathcal{Q} \subseteq \mathcal{O} \times [H+1]$ and inputs $\mathcal{AO}$.

An important class of policies for RDPs are the regular policies. Given an RDP $\mathbf{R}$, a policy $\pi : \mathcal{H} \to \Delta(\mathcal{A})$ is called *regular* if $\pi(h_1) = \pi(h_2)$ whenever $\bar{\tau}(h_1) = \bar{\tau}(h_2)$, for all $h_1, h_2 \in \mathcal{H}$. Let $\Pi_{\mathbf{R}}$ denote the set of regular policies for $\mathbf{R}$. Regular policies exhibit powerful properties. First, under a regular policy, suffixes have the same probability of being generated for histories that map to the same RDP state. Second, there exists at least one optimal policy that is regular.

**Proposition 1.** *Consider an RDP $\mathbf{R}$, a regular policy $\pi \in \Pi_{\mathbf{R}}$ and two histories $h_1$ and $h_2$ in $\mathcal{H}_t$, $t \in [H]$, such that $\bar{\tau}(h_1) = \bar{\tau}(h_2)$. For each suffix $e_{t+1:H} \in \mathcal{E}_{H-t-1}$, the probability of generating $e_{t+1:H}$ is the same for $h_1$ and $h_2$, i.e. $\mathbb{P}(e_{t+1:H} \mid h_1, \pi, \mathbf{R}) = \mathbb{P}(e_{t+1:H} \mid h_2, \pi, \mathbf{R})$.*

**Proposition 2.** *Each RDP $\mathbf{R}$ has at least one optimal policy $\pi^* \in \Pi_{\mathbf{R}}$.*

Due to Proposition 2, when solving an RDP $\mathbf{R}$, we can restrict our search to the set of regular policies $\Pi_{\mathbf{R}}$. A regular policy can be compactly defined as $\pi : \mathcal{Q} \to \Delta(\mathcal{A})$, where $\pi(q_0) = \mathbb{1}_{a_\perp}$ always selects the dummy action $a_\perp$, and its value function as $V^\pi : [H] \times \mathcal{Q} \to \mathbb{R}$.

Next, we define occupancy measures for RDPs. Given a regular policy $\pi : \mathcal{Q} \to \Delta(\mathcal{A})$ and $t \in [H]$, let $d_t^\pi \in \Delta(\mathcal{Q}_t \times \mathcal{A}\mathcal{O})$ be the induced probability distribution over states in $\mathcal{Q}_t$ and input symbols in $\mathcal{A}\mathcal{O}$, recursively defined as $d_0^\pi(q_0, a_0 o_0) := \theta_{\mathsf{o}}(q_0, a_0, o_0)$ and

$$d_t^\pi(q_t, a_t o_t) := \sum_{(q, ao) \in \tau^{-1}(q_t)} d_{t-1}^\pi(q, ao)\, \pi(q_t, a_t)\, \theta_{\mathsf{o}}(q_t, a_t, o_t) \quad t > 0.$$

We also overload the notation by writing $d_t^\pi(q_t, a_t) = \sum_{o \in \mathcal{O}} d^\pi(q_t, a_t o)$. Of particular interest is the occupancy distribution $d_t^* := d_t^{\pi^*}$, associated with an optimal policy $\pi^*$.

**Example 1** (The cookie domain [28])**.** The *cookie domain* (Figure 1a) has three rooms connected by a hallway. The agent (purple triangle) can move in the four cardinal directions. When pressing a button in the orange room, a cookie randomly appears in either the green or the blue room. The agent receives a reward of $+1$ for eating the cookie and it may then press the button again. This domain is partially observable since the agent can only see what is in the room that it currently occupies (Figure 1b). The cookie domain can be modelled as an episodic RDP with states $\mathcal{Q} = [H+1] \times \mathcal{O} \times \mathcal{U}$, with $\mathcal{U} = \{u_1, u_2, u_3, u_4\}$. The value of $\mathcal{U}$ is $u_1$ when the button has *not* been pressed yet (or not pressed since the last cookie was eaten). The value is $u_2$ when the button has been pressed, but the agent has not visited the blue or green room yet. In this state, the environment has a $50\%$ probability of generating the observation of a cookie when the agent enters either room for the first time. If the agent visits the green room and finds no cookie, the value becomes $u_3$, meaning that the cookie is in the blue room. The meaning of $u_4$ is dual to that of $u_3$.

## 2.2 Offline RL in RDPs

We are now ready to formally present the offline RL problem in episodic RDPs. Assume that we have access to a batch dataset $\mathcal{D}$ collected through interacting with an unknown (but fixed) episodic RDP $\mathbf{R}$ using a regular *behavior* policy $\pi^{\mathsf{b}}$. We assume that $\mathcal{D}$ comprises $N$ episodes, where the $k$-th episode is of the form $e_{0:H}^k = a_0^k r_0^k o_0^k \cdots a_H^k r_H^k o_H^k$, where $q_0^k = q_0$ and where, for each $t \in [H]$,

$$a_t^k \sim \pi^{\mathsf{b}}(q_t^k), \quad r_t^k \sim \theta_{\mathsf{r}}(q_t^k, a_t^k), \quad o_t^k \sim \theta_{\mathsf{o}}(q_t^k, a_t^k), \quad q_{t+1}^k = \tau(q_t^k, a_t^k o_t^k).$$

The goal is to compute a near-optimal policy $\widehat{\pi}$ using the dataset $\mathcal{D}$ (and without further exploration). More precisely, for a pre-specified accuracy $\varepsilon \in (0, H]$, we aim to find an $\varepsilon$-optimal policy $\widehat{\pi}$, using the smallest dataset $\mathcal{D}$ possible. A policy $\widehat{\pi}$ is $\varepsilon$-optimal iff $\mathbb{E}_{h_0}[V_0^*(h_0) - V_0^{\widehat{\pi}}(h_0)] \leq \varepsilon$, where $h_0 = a_\perp o_0$, for some random $o_0 \in \mathcal{O}$.

By virtue of Proposition 2, one may expect that it is sufficient to search for regular $\varepsilon$-optimal policies, which is indeed the case. In order to learn an $\varepsilon$-optimal policy from $\mathcal{D}$, some assumption is necessary regarding the policy $\pi^{\mathsf{b}}$ that was used to collect the episodes. Let $d_t^{\mathsf{b}} := d_t^{\pi^{\mathsf{b}}}$ be the occupancy distribution of $\pi^{\mathsf{b}}$. The following assumption requires that the behavior policy assigns a positive probability to all actions, which ensures that $\pi^{\mathsf{b}}$ explores the entire RDP.

**Assumption 1.** $\min_{t \in [H], q \in \mathcal{Q}_t, a \in \mathcal{A}} d_t^{\mathsf{b}}(q, a) > 0.$

Assumption 1 is only needed by Theorem 6, which reconstructs the full unknown RDP. Theorem 8, instead, relies on a weaker assumption that can be expressed with the coefficient introduced in Definition 1.

The second assumption we require concerns the richness of $\pi^{\mathsf{b}}$ and its capability to allow us to distinguish the various RDP states. This is perfectly captured by notions of *distiguishability* arising in automata theory, such as in Balle et al. [10]. We apply these concepts in our context, where such discrete distributions are generated from an RDP and a policy. Consider a minimal RDP $\mathbf{R}$ with states $\mathcal{Q} = \cup_{t \in [H+1]} \mathcal{Q}_t$. Given some policy $\pi$, at each timestep $t \in [H]$, every RDP state $q \in \mathcal{Q}_t$ defines a unique probability distribution over the episode suffixes $\mathcal{E}_{H-t} = (\mathcal{A}\mathcal{R}\mathcal{O})^{H-t+1}$. Then, the states in each $\mathcal{Q}_t$ can be compared through the probability distributions they induce over $\mathcal{E}_{H-t}$. Consider any $L = \{L_\ell\}_{\ell=0}^H$, where each $L_\ell$ is a metric over $\Delta(\mathcal{E}_\ell)$. We define the *L-distinguishability* of $\mathbf{R}$ and $\pi$ as the maximum $\mu_0$ such that, for any $t \in [H]$ and any two distinct $q, q' \in \mathcal{Q}_t$, the probability distributions over suffix traces $e_{t:H} \in \mathcal{E}_\ell$ from the two states satisfy

$$L_{H-t}(\mathbb{P}(e_{t:H} \mid q_t = q, \pi), \mathbb{P}(e_{t:H} \mid q_t = q', \pi)) \geq \mu_0 .$$

We will often omit the remaining length of the episode $\ell = H - t$ from $L_\ell$ and simply write $L$. We consider the $L_\infty^{\mathsf{p}}$-distinguishability, instantiating the definition above with the metric $L_\infty^{\mathsf{p}}(p_1, p_2) = \max_{u \in [\ell], e \in \mathcal{E}_u} |p_1(e*) - p_2(e*)|$, where $p_i(e*)$ represents the probability of the trace prefix $e \in \mathcal{E}_u$, followed by any trace $e' \in \mathcal{E}_{\ell-u-1}$. The $L_1^{\mathsf{p}}$-distinguishability is defined analogously using $L_1^{\mathsf{p}}(p_1, p_2) = \sum_{u \in [\ell], e \in \mathcal{E}_u} |p_1(e*) - p_2(e*)|$. We can now require a positive distinguishability with our second assumption.

**Assumption 2.** The behavior policy $\pi^{\mathsf{b}}$ has $L_\infty^{\mathsf{p}}$-distinguishability of at least $\mu_0 > 0$.

Finally, in order to capture the mismatch in occupancy measure between the optimal policy and the behavior policy, we introduce a key quantity called *single-policy RDP concentrability coefficient*, which extends the single-policy concentrability coefficient in MDPs to RDPs:

**Definition 1.** The *single-policy RDP concentrability coefficient* of an RDP $\mathbf{R}$ with episode horizon $H$ and with respect to a policy $\pi^{\mathsf{b}}$ is defined as:

$$C_{\mathbf{R}}^* = \max_{t \in [H], q \in \mathcal{Q}_t, ao \in \mathcal{A}\mathcal{O}} \frac{d_t^*(q, ao)}{d_t^{\mathsf{b}}(q, ao)} . \tag{2}$$

The concentrability coefficient in Definition 1 resembles the notions of concentrability in MDPs (e.g., [14, 15]). It should be stressed, however, that those in MDPs are defined in terms of observation-action pairs $(o, a)$, whereas $C_{\mathbf{R}}^*$ is defined in terms of *hidden* RDP states and actions-observations, $(q, ao)$. It is worth remarking that $C_{\mathbf{R}}^*$ could be equivalently defined in terms of state-action pairs $(q, a)$. Finally, in the special case where the RDP is Markovian – in which case it coincides with an episodic MDP – we have $\mathcal{Q} \subseteq \mathcal{O} \times [H+1]$ and $C_{\mathbf{R}}^*$ coincides with the standard single-policy concentrability coefficient for MDPs in [15]. This fact is shown in the proof of Corollary 17.

## 3 `RegORL`: Learning an Episodic RDP

In this section we present an algorithm for learning the transition function of an RDP $\mathbf{R}$ from a dataset $\mathcal{D}$ of episodes generated by a regular behavior policy $\pi^{\mathsf{b}}$. To simplify the presentation, we treat $\mathcal{D}$ as a multiset of traces in $\mathcal{E}_H$. The learning agent only has access to the non-Markovian traces

---

**Function** ADACT–H($\mathcal{D}$, $\delta$)

---

**Input:** Dataset $\mathcal{D}$ containing $N$ traces in $\mathcal{E}_H$, failure probability $0 < \delta < 1$
**Output:** Set $\mathcal{Q}$ of RDP states, transition function $\tau : \mathcal{Q} \times \mathcal{AO} \rightarrow \mathcal{Q}$

---

1  $\mathcal{Q}_0 \leftarrow \{q_0\}, \mathcal{X}(q_0) \leftarrow \mathcal{D}$                                    // initial state
2  **for** $t = 0, \dots, H$ **do**
3  $\quad$ $\mathcal{Q}_{c,t+1} \leftarrow \{qao \mid q \in \mathcal{Q}_t, ao \in \mathcal{AO}\}$                 // get candidate states
4  $\quad$ **foreach** $qao \in \mathcal{Q}_{c,t+1}$ **do** $\mathcal{X}(qao) \leftarrow \{e_{t+1:H} \mid aroe_{t+1:H} \in \mathcal{X}(q)\}$    // compute suffixes
5  $\quad$ $q_m a_m o_m \leftarrow \arg\max_{qao \in \mathcal{Q}_{c,t+1}} |\mathcal{X}(qao)|$                       // most common candidate
6  $\quad$ $\mathcal{Q}_{t+1} \leftarrow \{q_m a_m o_m\}, \tau(q_m, a_m o_m) = q_m a_m o_m$                        // promote candidate
7  $\quad$ $\mathcal{Q}_{c,t+1} \leftarrow \mathcal{Q}_{c,t+1} \setminus \{q_m a_m o_m\}$                          // remove from candidate states
8  $\quad$ **for** $qao \in \mathcal{Q}_{c,t+1}$ **do**
9  $\quad\quad$ $Similar \leftarrow \{q' \in \mathcal{Q}_{t+1} \mid \text{not } \text{TESTDISTINCT}(t, \mathcal{X}(qao), \mathcal{X}(q'), \delta)\}$    // confidence test
10 $\quad\quad$ **if** $Similar = \emptyset$ **then** $\mathcal{Q}_{t+1} \leftarrow \mathcal{Q}_{t+1} \cup \{qao\}, \tau(q, ao) = qao$    // promote candidate
11 $\quad\quad$ **else** $q' \leftarrow$ element in $Similar, \tau(q, ao) = q', \mathcal{X}(q') \leftarrow \mathcal{X}(q') \cup \mathcal{X}(qao)$    // merge states
12 $\quad$ **end**
13 **end**
14 **return** $\mathcal{Q}_0 \cup \dots \cup \mathcal{Q}_{H+1}, \tau$
15 **Function** TESTDISTINCT($t$, $\mathcal{X}_1$, $\mathcal{X}_2$, $\delta$)
16 $\quad$ **return** $L^p_\infty(\mathcal{X}_1, \mathcal{X}_2) \geq \sqrt{2 \log(8(ARO)^{H-t}/\delta)/\min(|\mathcal{X}_1|, |\mathcal{X}_2|)}$

---

in $\mathcal{D}$, and needs prior knowledge of $\mathcal{A}$, $\mathcal{R}$ and $\mathcal{O}$, but no prior knowledge of $\pi^b$ and $\mathbf{R}$. Our algorithm is an adaptation of ADACT [10] to episodic RDPs, and we thus refer to the algorithm as ADACT–H.

The intuition behind ADACT–H is that due to Proposition 1, two histories $h_1$ and $h_2$ should map to the same RDP state if they induce the same probability distribution on suffixes. ADACT–H starts by adding an initial RDP state $q_0$ to $\mathcal{Q}_0$, whose suffixes are the full traces in $\mathcal{D}$ (line 1). The algorithm then iteratively constructs the state sets $\mathcal{Q}_1, \dots, \mathcal{Q}_{H+1}$. In each iteration $t \in [H]$, ADACT–H creates a set of candidate states $\mathcal{Q}_{c,t+1}$ by extending all states in $\mathcal{Q}_t$ with symbols in $\mathcal{AO}$ (line 3). We use $qao$ to simultaneously refer to a candidate state and its state-symbol prefix $(q, ao)$. We associate each candidate state $qao$ with a multiset of suffixes $\mathcal{X}(qao)$, i.e. traces in $\mathcal{E}_{H-t-1}$, obtained by selecting all suffixes in $\mathcal{X}(q)$ that start with action $a$ and observation $o$ (line 4).

Next, ADACT–H finds the candidate state whose suffix multiset has maximum cardinality, and promotes this candidate to $\mathcal{Q}_{t+1}$ by defining the transition function $\tau$ accordingly (lines 5-7). The algorithm then iterates over each remaining candidate state $qao \in \mathcal{Q}_{c,t+1}$, comparing the distribution on suffixes in $\mathcal{X}(qao)$ to those of states in $\mathcal{Q}_{t+1}$ (line 9). If the suffix distribution is different from that of each state in $\mathcal{Q}_{t+1}$, $qao$ is promoted to $\mathcal{Q}_{t+1}$ (line 10), else $qao$ is merged with a state $q' \in \mathcal{Q}_{t+1}$ that has a similar suffix distribution (line 11). Finally, ADACT–H returns the set of RDP states $\mathcal{Q}$ and the associated transition function $\tau$. The function TESTDISTINCT compares two multisets $\mathcal{X}_1$ and $\mathcal{X}_2$ of traces in $\mathcal{E}_{H-t-1}$ using the metric $L^p_\infty$. For $i \in \{1, 2\}$ and each trace $e \in \mathcal{E}_{H-t-1}$, let $\widehat{p}_i(e) = \sum_{x \in \mathcal{X}_i} \mathbb{I}(x = e)/|\mathcal{X}_i|$ be the empirical estimate of $p_i$, as the proportion of elements in $\mathcal{X}_i$ equal to $e$. TESTDISTINCT compares $L^p_\infty(\mathcal{X}_1, \mathcal{X}_2) := L^p_\infty(\widehat{p}_1, \widehat{p}_2)$ to a confidence threshold.

**Markov transformation.** We are now ready to connect the RDP learning phase with the MDP learning phase. RDPs do not respect the Markov property over their observations and rewards, if automaton states remain hidden. However, we can use the reconstructed transition function $\tau$ returned by ADACT–H, extended over histories as $\bar{\tau} : \mathcal{H} \rightarrow \mathcal{Q}$, to recover the Markov property. In what follows we formalize the notion of Markov transformation and the properties that its outputs satisfy.

**Definition 2.** Let $e_{0:H} \in \mathcal{E}_H$ be an episode collected from an RDP $\mathbf{R}$ and a policy $\pi^b$ that is regular in $\mathbf{R}$. The *Markov transformation* of $e_H$ with respect to $\mathbf{R}$ is the episode constructed as $a_0 r_0 q_1 \dots a_H r_H q_{H+1}$, where $q_{t+1} = \bar{\tau}(h_t)$ and $h_t = a_0 o_0 \cdots a_t o_t$, $t \in [H]$. The Markov transformation of a dataset $\mathcal{D}$ is the Markov transformation of all the episodes it contains.

A Markov transformation discards all observations from $\mathcal{D}$ and replaces them with RDP states output by $\bar{\tau}$. The dataset so constructed can be seen as generated from an MDP, which we define next.

**Definition 3.** The episodic MDP *associated to* an episodic RDP $\mathbf{R}$ is $\mathbf{M_R} = \langle \mathcal{Q}, \mathcal{A}, \mathcal{R}, T, \theta_r, H \rangle$, where $T(q, a, q') = \sum_{o \in \mathcal{O}} \mathbb{I}(q' = \tau(q, ao)) \theta_o(q, a, o)$ for each $(q, a, q') \in \mathcal{Q} \times \mathcal{A} \times \mathcal{Q}$.

The associated MDP in Definition 3 is the decision process that corresponds to the Markov transformation of Definition 2, i.e. any episode produced with the Markov transformation can be equivalently seen as being generated from the associated MDP, in the sense of the following proposition.

**Proposition 3.** *Let $e_{0:H}$ be an episode sampled from an episodic RDP $\mathbf{R}$ under a regular policy $\pi \in \Pi_{\mathbf{R}}$, with $\pi(h, a) = \pi_r(\bar{\tau}(h), a)$. If $e'_H$ is the Markov transformation of $e_H$ with respect to $\mathbf{R}$, then $\mathbb{P}(e'_H \mid \mathbf{R}, \pi) = \mathbb{P}(e'_H \mid \mathbf{M}_{\mathbf{R}}, \pi_r)$, where $\mathbf{M}_{\mathbf{R}}$ is the MDP associated to $\mathbf{R}$.*

Rewards are not affected by the Markov transformation, only observations, implying the following.

**Proposition 4.** *Let $\pi \in \Pi_{\mathbf{R}}$ be a regular policy in $\mathbf{R}$ such that $\pi(h, a) = \pi_r(\bar{\tau}(h), a)$. Then $\mathbb{E}[V_{0,\mathbf{R}}^{\pi}] = \mathbb{E}[V_{0,\mathbf{M}_{\mathbf{R}}}^{\pi_r}]$, where $V_{0,\mathbf{R}}^{\pi}$ and $V_{0,\mathbf{M}_{\mathbf{R}}}^{\pi_r}$ are the values in the respective decision process, and $\mathbb{E}[V_{0,\mathbf{R}}^{*}] = \mathbb{E}[V_{0,\mathbf{M}_{\mathbf{R}}}^{*}]$, where expectations are with respect to randomness in $o_0$.*

**Corollary 5.** *Given $\varepsilon \in (0, H]$, if $\pi_r : \mathcal{Q} \to \Delta(\mathcal{A})$ is an $\varepsilon$-optimal policy of $\mathbf{M}_{\mathbf{R}}$, the MDP associated to some RDP $\mathbf{R}$, then, $\pi(h, a) = \pi_r(\bar{\tau}(h), a)$ is $\varepsilon$-optimal in $\mathbf{R}$.*

In summary, from Proposition 3, if $\mathcal{D}_m$ is the Markov transformation of a dataset $\mathcal{D}$ with respect to an RDP $\mathbf{R}$, then, $\mathcal{D}_m$ can be seen as being generated from the associated MDP $\mathbf{M}_{\mathbf{R}}$. Hence, any offline RL algorithm for MDPs can be used for learning in $\mathcal{D}_m$. Moreover, according to Corollary 5, any solution for $\mathbf{M}_{\mathbf{R}}$ can be translated via $\bar{\tau}$ into a policy for the original RDP, with the same guarantees.

**Complete algorithm.** The complete procedure is illustrated in Algorithm 1. Initially, the input dataset $\mathcal{D}$ is separated in two halves. The first portion is used for learning the transition function of the unknown RDP with ADACT–H (Section 3). If an upper bound $\overline{Q}$ on $|\mathcal{Q}|$ is available, it can optionally be provided to compute a more appropriate failure parameter for ADACT–H. If not available, we adopt the upper bound of $2(AO)^H$ states, which is valid for any instance, due to histories having finite length. As we will see in Theorem 6, this would only contribute linearly in $H$ to the required dataset size. The output function computed by ADACT–H is then used to compute a Markov transformation of the second phase, as specified in Definition 2. The resulting dataset, now Markovian, can be passed to a generic offline RL algorithm, which we represent with the function OFFLINERL$(\mathcal{D}, \varepsilon, \delta)$. In Appendix D, we instantiate it for a specific state-of-the-art offline RL algorithm.

---

**Algorithm 1:** Full procedure (`RegORL`)

---

**Input:** Dataset $\mathcal{D}$, accuracy $\varepsilon \in (0, H]$, failure probability $0 < \delta < 1$, (optionally) upper bound $\overline{Q}$ on $|\mathcal{Q}|$
**Output:** Policy $\hat{\pi} : \mathcal{H} \to \Delta(\mathcal{A})$

1  $\mathcal{D}_1, \mathcal{D}_2 \leftarrow$ separate $\mathcal{D}$ into two datasets of the same size
2  $\mathcal{Q}, \tau \leftarrow$ ADACT–H$(\mathcal{D}_1, \delta/(4AO\overline{Q}))$, where $\overline{Q} = 2(AO)^H$ if not provided
3  $\mathcal{D}'_2 \leftarrow$ Markov transformation of $\mathcal{D}_2$ with respect to $\bar{\tau}$ as in Definition 2
4  $\hat{\pi}_m \leftarrow$ OFFLINERL$(\mathcal{D}'_2, \varepsilon, \delta/2)$
5  **return** $\hat{\pi} : h \mapsto \hat{\pi}_m(\bar{\tau}(h))$

---

## 4 Theoretical Guarantees

We now turn to theoretical performance guarantees of `RegORL`. Our main performance result is a sample complexity bound in Theorem 7, ensuring that, for any accuracy $\varepsilon \in (0, H]$, `RegORL` finds an $\varepsilon$-optimal policy. We also report a sample complexity bound for ADACT–H in Theorem 6, and an alternative bound in Theorem 8. In comparison, the sample complexity bound for ADACT [10] is

$$\widetilde{\mathcal{O}} \left( \frac{Q^4 A^2 O^2 H^5 \log(1/\delta)}{\varepsilon^2} \max \left\{ \frac{1}{\mu_0^2}, \frac{H^4 O^2 A^2}{\varepsilon^4} \right\} \right).$$

We achieve a tighter bound by using Bernstein's inequalities and exploiting the finiteness of histories.

**Theorem 6.** *Consider a dataset $\mathcal{D}$ of episodes sampled from an RDP $\mathbf{R}$ and a regular policy $\pi^b \in \Pi_{\mathbf{R}}$. With probability $1 - \delta$, the output of ADACT–H$(\mathcal{D}, \delta/(2QAO))$ is the transition function of the minimal RDP equivalent to $\mathbf{R}$, provided that $|\mathcal{D}| \geq N_\delta$, where*

$$N_\delta := \frac{21 \log(8QAO/\delta)}{d_{\min}^b \mu_0} \sqrt{H \log(2ARO)} \in \widetilde{\mathcal{O}} \left( \frac{\sqrt{H}}{d_{\min}^b \mu_0} \right),$$

$d_{\min}^{\mathsf{b}} := \min\{d_t^{\mathsf{b}}(q, ao) \mid t \in [H], q \in \mathcal{Q}_t, ao \in \mathcal{AO}, d_t^{\mathsf{b}}(q, ao) > 0\}$ *is the minimal occupancy distribution, and $\mu_0$ is the $L_\infty^{\mathsf{p}}$-distinguishability.*

The proof appears in Appendix C.2. Theorem 6 tells us that the sample complexity of ADACT–H, to return a minimal RDP, is inversely proportional to $\mu_0$, the $L_\infty^{\mathsf{p}}$-distinguishability of $\mathbf{R}$ and $\pi^{\mathsf{b}}$, and the minimal occupancy $d_{\min}^{\mathsf{b}}$. Note that $d_{\min}^{\mathsf{b}} \leq 1/(QOA)$. The bound also depends on $Q$, the number of RDP states, implicitly through $d_{\min}^{\mathsf{b}}$ and explicitly via a logarithmic term. In the absence of prior knowledge of $Q$, one may use in the argument of Algorithm 1 the worst-case upper bound $\overline{Q} = 2(AO)^H$. The sample complexity would then have an additional linear term in $H$, since $\overline{Q}$ is only used in the logarithmic term to set the appropriate value of $\delta$. However, this will not impact the value of the $d_{\min}^{\mathsf{b}}$ term.

Theorem 6 is a sample complexity guarantee for the first phase of the algorithm, which learns $\tau$, the structure of the minimal RDP that is equivalent to the underlying RDP. If $\delta$ is the desired failure probability of the complete algorithm, RegORL executes ADACT–H so that its success probability is at least $1 - \delta/2$. This means that with the same probability, $\mathcal{D}_2'$ is an MDP dataset with the properties listed in Section 3. As a consequence, provided that OFFLINERL is some generic $(\varepsilon, \delta/2)$-PAC offline RL algorithm for MDPs, the output of RegORL is an $\varepsilon$-optimal policy with probability $1 - \delta$.

**Theorem 7.** *Consider a dataset $\mathcal{D}$ of episodes sampled from an RDP $\mathbf{R}$ and a regular policy $\pi^{\mathsf{b}} \in \Pi_\mathbf{R}$. For any $\varepsilon \in (0, H]$ and $0 < \delta < 1$, if OFFLINERL is an $(\varepsilon, \delta/2)$-PAC offline algorithm for MDPs with sample complexity $N_{\mathsf{m}}$, then, the output of RegORL$(\mathcal{D}, \varepsilon, \delta)$ is an $\varepsilon$-optimal policy in $\mathbf{R}$, with probability at least $1 - \delta$, provided that $|\mathcal{D}| \geq 2 \max\{N_{\delta/2}, N_{\mathsf{m}}\}$.*

As we can see, the sample complexity requirement separates for the two phases. While $N_{\delta/2}$ is due to the RDP learning component, defined in Theorem 6, the quantity $N_{\mathsf{m}}$ completely depends on the offline RL algorithm for MDPs that is adopted. Among other terms, the performance guarantees of offline algorithms can often be characterized through the single-policy concentrability for MDPs $C^*$. However, since states become observations in the associated MDP, due to the properties of Proposition 3, $C^*$ coincides with $C_\mathbf{R}^*$, the RDP single-policy concentrability of Definition 1.

In Appendix D, we demonstrate a specific instantiation of RegORL with an off-the-shelf offline RL algorithm from the literature by Li et al. [16]. This yields the following requirement for $N_{\mathsf{m}}$:

$$N_{\mathsf{m}} \geq \frac{c\, H^3 Q C_\mathbf{R}^* \log \frac{2H N_{\mathsf{m}}}{\delta}}{\varepsilon^2},$$

for a constant $c > 0$.

To eliminate the dependence that Theorem 6 has on $d_{\min}^{\mathsf{b}}$, we develop a variant of ADACT–H which does not learn a complete RDP. Rather, it only reconstructs a subset of states that are likely under the behavior policy. The algorithm, which we call ADACT–H–A (with 'A' standing for "approximation"), is defined in Appendix C.3. Theorem 8 is an upper bound on the sample complexity of ADACT–H–A that takes the accuracy $\varepsilon$ as input and returns the transition function of an $\varepsilon/2$-approximate RDP $\mathbf{R}'$, whose optimal policy is $\varepsilon/2$-optimal for the original RDP $\mathbf{R}$. By performing a Markov transformation for $\mathbf{R}'$ and using an $(\varepsilon/2, \delta/2)$-PAC offline algorithm for MDPs, we can compute an $\varepsilon$-optimal policy for $\mathbf{R}$. The total sample complexity can be combined in the same way as in Theorem 7.

**Theorem 8.** *Consider a dataset $\mathcal{D}$ of episodes sampled from an RDP $\mathbf{R}$ and a regular policy $\pi^{\mathsf{b}} \in \Pi_\mathbf{R}$. With probability $1 - \delta$, the output of ADACT–H–A, called with $\mathcal{D}$, $\delta/(2QAO)$ and $\varepsilon \in (0, H]$ in input, is the transition function of an $\varepsilon/2$-approximate RDP $\mathbf{R}'$, provided that $|\mathcal{D}| \geq N_\delta'$, where*

$$N_\delta' := \frac{504 H Q A O C_{\mathbf{R}'}^* \log(16QAO/\delta)}{\varepsilon\, \mu_0} \sqrt{H \log(2ARO)} \in \widetilde{\mathcal{O}}\left(\frac{H^{3/2} QAO C_{\mathbf{R}'}^*}{\varepsilon\, \mu_0}\right).$$

This theorem does not rely on Assumption 1, because a finite $C_\mathbf{R}^*$ suffices.

## 5  Sample Complexity Lower Bound

The main result of this section is Theorem 9, a sample complexity lower bound for offline RL in RDPs. It shows that the dataset size required by any RL algorithm scales with the relevant parameters.

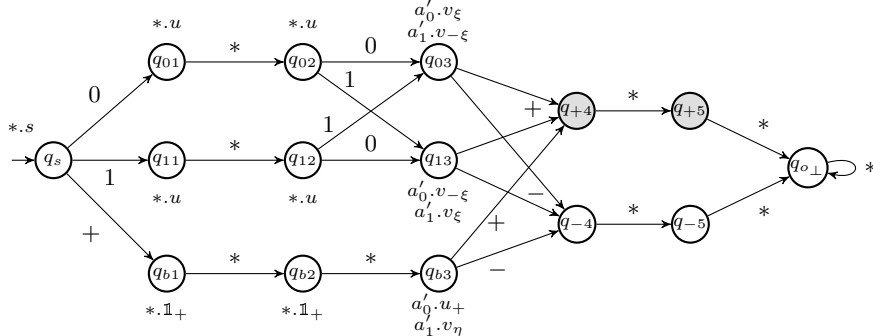

Figure 2: One episodic RDP instance $\mathbf{R}_{101,1} \in \mathbb{R}(L, H, \xi, \eta)$, associated to the parity function $f_{101}$, with code 101, and the optimal arm $a'_1$. The length is $L = |101| = 3$, the horizon $H = 5$, the noise parameter $\xi$ and the bandit bonus parameter is $\eta$. The transition function only depends on the observations, not the actions. The output distributions are: $u = \text{unif}\{0, 1\}$, $u_+ = \text{unif}\{+, -\}$, $v_\alpha(+) = (1 + \alpha)/2$, $v_\alpha(-) = (1 - \alpha)/2$. The star denotes any symbol. If the label of a state $q$ is $a.d$, then the observation function is $\theta_o(q, a) = d$. Refer to Appendix E for details.

**Theorem 9.** *For any $(C^*_{\mathbf{R}}, H, \varepsilon, \mu_0)$ satisfying $C^*_{\mathbf{R}} \geq 2$, $H \geq 2$ and $\varepsilon \leq H\mu_0/64$, there exists an RDP with horizon $H$, $L^p_1$-distinguishability $\mu_0$ and a regular behavior policy $\pi^b$ with RDP single-policy concentrability $C^*_{\mathbf{R}}$, such that if $\mathcal{D}$ has been generated using $\pi^b$ and $\mathbf{R}$, and*

$$|\mathcal{D}| \notin \Omega\left(\frac{H}{\mu_0} + \frac{C^*_{\mathbf{R}} H^2}{\varepsilon^2}\right) \tag{3}$$

*then, for any algorithm $\mathfrak{A} : \mathcal{D} \mapsto \widehat{\pi}$ returning non-Markov deterministic policies, the probability that $\widehat{\pi}$ is not $\varepsilon$-optimal is at least $1/4$.*

The proof relies on worst-case RDP instances that carefully combine two-armed bandits with noisy parity functions. This last component allows to capture the difficulty of learning in presence of temporal dependencies. Figure 2 shows an RDP in this class. At the beginning of each episode, the observation causes a transition towards either the bandit component (bottom branch) or the noisy parity function (top branches). Acting optimally in the two parity branches requires to predict the output of a parity function, which depends on some unknown binary code (of length 3, in the example). The first term in Theorem 9 is due to this component, because the code scales linearly with $H$, or $Q$, while the amount of information revealed about the code is controlled by $\mu_0$. The second term is caused by the required optimality in the bandit.

Differently from this lower bound, the parameter $\mu_0$, appearing in the upper bounds of Theorems 6 and 8, is a $L^p_\infty$-distinguishability. However, the two are related, since $L^p_1(q, q') \geq L^p_\infty(q, q')$. Intuitively, the $L^p_1$-distinguishability accounts for all the information that is available as differences in episode probabilities. The $L^p_\infty$-distinguishability, on the other hand, quantifies the maximum the difference in probability associated to specific suffixes. This is the information used by the algorithm and the one appearing in the two upper bounds.

## 6 Conclusion

In this paper we propose an algorithm for Offline RL in episodic Regular Decision Processes. Our algorithm exploits automata learning techniques to reduce the problem of RL in RDPs, in which observations and rewards are non-Markovian, into standard offline RL for MDPs. We provide the first high-probability sample complexity guarantees for this setting, as well as a lower bound that shows how its complexity relates to the parameters that characterize the decision process and the behavior policy. We identify the RDP single-policy concentrability as an analogous quantity to the one used for MDPs in the literature. Our sample complexity upper bound depends on the $L^p_\infty$-distinguishability of the behavior policy. As a future work, we plan to investigate if any milder notion of distinguishability also suffices. This is motivated by our lower bound which only involves the $L^p_1$-distinguishability over the same policy. Finally, our results have strong implications for online learning in RDPs, which is a relevant setting to be explored.

## Acknowledgments

Roberto Cipollone is partially supported by the EU H2020 project AIPlan4EU (No. 101016442), the ERC-ADG White-Mech (No. 834228), the EU ICT-48 2020 project TAILOR (No. 952215), the PRIN project RIPER (No. 20203FFYLK), and the PNRR MUR project FAIR (No. PE0000013). Anders Jonsson is partially supported by the EU ICT-48 2020 project TAILOR (No. 952215), AGAUR SGR, and the Spanish grant PID2019-108141GB-I00. Alessandro Ronca is partially supported by the ERC project WhiteMech (No. 834228), and the ERC project ARiAT (No. 852769). Mohammad Sadegh Talebi is partially supported by the Independent Research Fund Denmark, grant number 1026-00397B.

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

# Appendices

# A   Extended Discussion of Related Work

Some of the related work mentioned in the introduction requires a more extensive discussion, which we provide below.

## A.1   Offline RL in MDPs

There is a rich and growing literature on offline RL in MDPs; see, e.g., [11, 12, 13, 14, 15, 16, 17, 18, 19, 20]. A closely related line of work is off-policy learning in MDPs [21, 22, 23]; we refer to the recent survey [24] for a discussion on how the various settings are related. For offline RL in MDPs, the papers cited above report learning algorithms with theoretical guarantees on their sample efficiency. The majority of these algorithms are designed based on the *pessimism principle*. While most literature focuses on tabular MDPs, the case of linear function approximation is discussed in some papers, e.g., [20]. In several settings, the presented algorithms are shown to be minimax optimal. For instance, in the case of tabular episodic MDPs, it is established in [16] that the optimal sample complexity depends on the size of state-space, episode length, as well as some notion of concentrability reflecting the distribution mismatch between the behavior and optimal policy.

## A.2   Online RL in RDPs

RDPs have been introduced in [3] as a formalism based on temporal logic. They admit an equivalent formulation in terms of automata, which is favoured in the context of RL. Several algorithms for *online* RL in RDPs exist [25, 26, 27] but complexity bounds are only given in [26] for the infinite-horizon discounted setting. This work [26] shows the correspondence between RDPs and Probabilistic Deterministic Finite Automata (PDFA), and it introduces the idea of using PDFA-learning techniques to learn RDPs. Their sample complexity bounds are not immediately comparable to ours, due to the different setting. Importantly, this algorithm uses the uniform policy for learning. So, the algorithm might be adapted to our setting only under the assumption that the behaviour policy is uniform. Even in this case, our bounds show an improved dependency on several key quantities. Furthermore, we provide a sample complexity lower bound, whereas their results are limited to showing that a dependency on the quantities occurring in their upper bounds is necessary.

The first RL algorithm for RDPs appears in [25] for the online discounted setting. It is automaton-based, and in particular, it learns the RDP in the form of a Mealy machine. The algorithm is shown in [26] to incur an exponential dependency on the length of the relevant histories. An algorithm that achieves performance guarantees building on the techniques from [26] but also integrating an effective exploration strategy is given in [27]. This work also introduces the idea of seeing the transition function of a PDFA as a Markov abstraction of the histories to be passed to an RL algorithm for MDPs, so as to employ it in a modular manner.

The algorithms in [28, 29, 30, 31] apply to RDPs even though they have not been developed specifically for RDPs. Toro Icarte et al. [28] present an RL algorithm for the subclass of POMDPs that have a finite set of reachable belief states. Such POMDPs can be captured by finite-state automata, and are in fact RDPs. Their algorithm is based on automata learning, but it does not come with an analysis of its performance guarantees. The RL techniques presented in [29, 30] for *feature MDPs* are in fact applicable to episodic RDPs. The techniques are based on suffix trees, rather than automata. There are cases when the size of the smallest suffix tree is exponential in the horizon, while an automaton of linear size exists—see section below on feature MDPs. Thus their techniques cannot yield optimal bounds for RDPs. Mahmud [31] introduces an RL algorithm for *Deterministic Markov Decision Models* (MDDs). Such MDDs are also automaton-based, and they are more general than RDPs since the automaton is only required to predict rewards. Thus their RL algorithm applies to RDPs. However, it is an algorithm without guarantees.

## A.3   Non-Markov Rewards and Reward Machines

MDPs with non-Markov rewards are a special case of NMDPs, where only rewards are non-Markovian. Namely, observed states satisfy the Markov property, while rewards may depend on the history of past states. The specific kind of non-Markov rewards considered in the literature amount to the subclass of RDPs where the automaton's state is only needed to predict the next reward—while the next observation (i.e., state) can be predicted from the last observation.

Non-Markov rewards can already be found in [2], where the reward function is specified in a temporal logic of the past. More recently, the setting has been revisited with so-called *reward machines* [44] as well as with temporal logics of the future on finite traces [64, 47]. A reward machine is a finite automaton (or transducer) used to specify a non-Markovian reward function. Reward machines have been introduced in [44] along with an RL algorithm that assumes the reward machine to be known. RL algorithms with unknown reward machine, or equivalently unknown temporal specification, are presented in [65, 45], with no performance guarantees. Reward machines have been generalised so as to predict observations as well [28], which makes them equivalent to RDPs—as mentioned above.

The first performance bounds for RL with reward machines have been recently established in [49]. The work shows regret bounds that take into account the structure provided by a reward machine, and hence improve over the bounds that one would obtain by naively adapting regret bounds for MDPs.

An automaton-based method for dealing with non-Markov sparse rewards is proposed in [48].

## A.4 State Representations

State representations are maps from histories to a finite state space. The map defined by the transition function of an RDP is a state representation. The studies on state representations [39, 40, 41, 42] focus on regret bounds for RL given a candidate set of state representations. While in our case the state representations are concretely defined by the class of finite-state automata, in their case they are arbitrary maps. This is a challenging setting, which does not allow for taking advantage of the properties of specific classes of state representations. The regret bounds in [39, 41, 42] are for finite sets of state representations, and they all show a linear dependency on the cardinality of the given set of state representations. In our case, the candidate state representations correspond to the set of automata with at most $Q = 2(AO)^H$ states and $AO$ input letters. Such a set contains at least $Q^{Q^{AO}}$ automata—the number of distinct transition functions. Thus, if we could instantiate their bounds in our setting, they would have an exponential dependency on the number $Q$ of RDP states, and hence a doubly-exponential dependency on the horizon $H$. We avoid this dependency, obtaining polynomial bounds in the mentioned quantities.

Nguyen et al. [40] consider the case of a countably infinite set of state representations, and present an algorithm whose regret bound does not show a dependency such as the one discussed above. Instead, they show a dependency on a quantity $K_0$, which admits several interpretations, including one based on the descriptional complexity of the candidate state representations. Thus, there may be a way to relate $K_0$ to the quantities we use in our bounds. However, the formal relationship between the two, if any, renders highly non-trivial, which prevents one to use their ideas in the case of RDPs. We believe establishing a fomal relationship between their model and RDPs is an interesting, yet challenging, topic for future work. Furthermore, it should be stressed that even if the relationship was clear and one could borrow ideas from [40], the resulting sample complexity bound would have to grow as $1/\varepsilon^3$ in view of their regret bound scaling as $T^{2/3}$. In contrast, our bounds achieve an optimal dependency of $1/\varepsilon^2$ on $\varepsilon$.

## A.5 Feature MDPs and General RL

Hutter [29] introduces *feature MDPs*, where histories are mapped to states by a feature map. It relates to our work since the map provided by the transition function of an RDP is a feature map. The concrete feature maps considered in [29] are based on U-Trees [66]. The idea is also revisited in [30] with Prediction Suffix Trees (PSTs) [67, 68]. Both U-Trees and PSTs are suffix trees. There are cases when their size is exponential in the horizon, while an automaton of linear size exists. For instance, in the case of a parity condition over the history. To see this, note that a suffix $x$ of a bit string $bx$ does not suffice to establish parity of $bx$. In fact, the parity of $0x$ is different from the parity of $1x$. Thus a suffix tree for parity must encode all suffixes, and hence it will have a number of leaves that is exponential in the maximum length of a relevant string—the horizon $H$ in the case of episodic RL.

Hutter [69] provides several formal characterisations of feature maps. All the characterisations are less restrictive than the one defined by an RDP. In particular, their most restrictive characterisation is given in their Equation (6). It only requires to preserve the ability to predict rewards, not observations—this is also adopted by Mahmud [31]. The states of our automata suffice to predict observations as well. It is unclear whether automata techniques can be used to learn directly abstractions that do not preserve the dynamics entirely.

Majeed and Hutter [55] study convergence of Q-learning when executed over the state space of an MDP that underlies a non-Markov decision processes. Such a state space corresponds to the state space of the RDP, but they do not consider the problem of learning the state space.

Lattimore et al. [43] consider General RL as the problem of RL when we are given a set of candidate NMDPs, rather than assuming the decision process to belong to a fixed class. Similarly to the works on state representations, it does not commit to specific classes of NMDPs, and their bounds have a linear dependency on the number of candidate models. As remarked above, in our setting, it amounts to an exponential dependency on the number of states of the candidate RDPs, and hence a doubly-exponential dependency on the horizon; we avoid such exponential dependencies.

### A.6 Learning PDFA

Our algorithms for learning an RDP borrow and improve over techniques for learning Probabilistic-Deterministic Finite Automata (PDFA). The first PAC learning algorithm for acyclic PDFA has been presented in [56], then followed by extensions and variants that can handle PDFA with cycles [57, 58, 10, 59, 60]. All bounds feature some variant of a *distinguishability parameter*, which we adopt in our bounds, properly adapting it to the offline RL setting. Our algorithm builds upon the state-of-the-art algorithm ADACT [10], and we derive bounds that are a substantial improvement over the ones that can be obtained from a straightforward application of any existing PDFA-learning algorithm to the offline RL setting.

## B  RDP properties

In this section we prove several properties of RDPs that are stated in Sections 2 and 3.

### B.1  RDPs and Regular Policies

In this section, we prove Propositions 1 and 2.

**Proposition 1.** *Consider an RDP $\mathbf{R}$, a regular policy $\pi \in \Pi_{\mathbf{R}}$ and two histories $h_1$ and $h_2$ in $\mathcal{H}_t$, $t \in [H]$, such that $\bar{\tau}(h_1) = \bar{\tau}(h_2)$. For each suffix $e_{t+1:H} \in \mathcal{E}_{H-t-1}$, the probability of generating $e_{t+1:H}$ is the same for $h_1$ and $h_2$, i.e. $\mathbb{P}(e_{t+1:H} \mid h_1, \pi, \mathbf{R}) = \mathbb{P}(e_{t+1:H} \mid h_2, \pi, \mathbf{R})$.*

*Proof.* By induction on $t$. For $t = H$, all histories in $\mathcal{H}_H$ generate the empty suffix in $(\mathcal{A}\mathcal{R}\mathcal{O})^0$ with probability 1 (the stop symbol is omitted). For $t < H$, the probability of generating a suffix $aroe_{t+2:H}$ is

$$\mathbb{P}(aroe_{t+2:H} \mid h_1, \pi, \mathbf{R}) = \pi(h_1, a) \cdot \mathbb{P}(r, o \mid \bar{\tau}(h_1), a, \mathbf{R}) \cdot \mathbb{P}(e_{t+2:H} \mid h_1 ao, \pi, \mathbf{R})$$
$$= \pi(h_2, a) \cdot \mathbb{P}(r, o \mid \bar{\tau}(h_2), a, \mathbf{R}) \cdot \mathbb{P}(e_{t+2:H} \mid h_2 ao, \pi, \mathbf{R}) = \mathbb{P}(aroe_{t+2:H} \mid h_2, \pi, \mathbf{R}),$$

where we have used the fact that $\pi$ is regular, $\bar{\tau}(h_1) = \bar{\tau}(h_2)$, $\bar{\tau}(h_1 ao) = \tau(\bar{\tau}(h_1), ao) = \tau(\bar{\tau}(h_2), ao) = \bar{\tau}(h_2 ao)$, and the induction hypothesis. $\square$

The following statement appears in the literature [3, Theorem 2], but the authors do not provide a complete proof, so for completeness we prove the statement here.

**Proposition 2.** *Each RDP $\mathbf{R}$ has at least one optimal policy $\pi^* \in \Pi_{\mathbf{R}}$.*

*Proof.* Given $\mathbf{R}$, consider any optimal policy $\pi^* : \mathcal{H} \to \Delta(\mathcal{A})$, not necessarily regular. We prove the statement by constructing a policy $\pi$ and showing by induction on $t \in [H]$ that $\pi$ is both optimal and regular. The base case is given by $t = H$. In this case, for an arbitrary $a \in \mathcal{A}$, define $\pi(h) := \mathbb{1}_a$ for each history $h \in \mathcal{H}_H$. Since $V_H^\pi(h) = 0$ by definition, $\pi$ is optimal for each history $h \in \mathcal{H}_H$, and regular since it always selects the same action.

For $t < H$, we first construct a new policy $\pi_{\mathsf{c}}$ which is the composition of policies $\pi^*$ and $\pi$. Concretely, for each history $h \in \mathcal{H}_u$ such that $u \leq t$, $\pi_{\mathsf{c}}(h) = \pi^*(h)$ acts according to $\pi^*$, while for each history $h \in \mathcal{H}_u$ such that $u > t$, $\pi_{\mathsf{c}}(h) = \pi(h)$ acts according to $\pi$. Clearly, $\pi_{\mathsf{c}}$ is an optimal policy for $\mathbf{R}$ since $\pi^*$ is optimal and since by induction, $\pi$ is optimal for histories in $\mathcal{H}_u$, $u > t$.

Consider a pair of histories $h_1$ and $h_2$ in $\mathcal{H}_t$ such that $\bar{\tau}(h_1) = \bar{\tau}(h_2)$ but $\pi_{\mathsf{c}}(h_1) \neq \pi_{\mathsf{c}}(h_2)$. Define $\pi(h_1) := \pi(h_2) := \pi_{\mathsf{c}}(h_1)$. Since the value function can be written as an expectation

over suffixes, due to Proposition 1 and the fact that $\pi$ is regular for histories in $\mathcal{H}_u$, $u > t$, we have $V_t^\pi(h_1) = V_t^\pi(h_2)$. Since $\pi_c$ is the same as $\pi$ for histories in $\mathcal{H}_u$, $u > t$, this implies $V_t^\pi(h_1) = V_t^{\pi_c}(h_1) \leq V_t^{\pi_c}(h_2)$ since $\pi_c$ is optimal for $h_2$. If we were to instead define $\pi(h_1) := \pi(h_2) := \pi_c(h_2)$, we would obtain $V_t^{\pi_c}(h_2) \leq V_t^{\pi_c}(h_1)$. The only possibility is $V_t^{\pi_c}(h_1) = V_t^{\pi_c}(h_2)$, which is the same value achieved by the policy $\pi$. Hence $\pi$ is optimal for $h_1$ and $h_2$.

We now repeat the same procedure for each pair of histories $h_1$ and $h_2$ in $\mathcal{H}_t$ such that $\bar{\tau}(h_1) = \bar{\tau}(h_2)$ but $\pi_c(h_1) \neq \pi_c(h_2)$. If necessary, we complete the definition of $\pi$ by copying the action choices of $\pi_c$. The resulting policy $\pi$ is optimal for each history $h \in \mathcal{H}_t$, and regular since it makes the same action choices for each pair of histories $h_1$ and $h_2$ in $h \in \mathcal{H}_t$ such that $\bar{\tau}(h_1) = \bar{\tau}(h_2)$. $\qquad\square$

## B.2 Markov Transformation

In this section, we verify the properties of the Markov transformation, which is the intermediate step that RegORL uses to recover the Markov property in the original dataset. This transformation has been formalized in Definition 2. We use $\mathcal{D}$ and $\mathcal{D}'$ to denote the original and the transformed datasets, respectively.

**Proposition 3.** *Let $e_{0:H}$ be an episode sampled from an episodic RDP $\mathbf{R}$ under a regular policy $\pi \in \Pi_\mathbf{R}$, with $\pi(h, a) = \pi_r(\bar{\tau}(h), a)$. If $e'_H$ is the Markov transformation of $e_H$ with respect to $\mathbf{R}$, then $\mathbb{P}(e'_H \mid \mathbf{R}, \pi) = \mathbb{P}(e'_H \mid \mathbf{M_R}, \pi_r)$, where $\mathbf{M_R}$ is the MDP associated to $\mathbf{R}$.*

*Proof.* For $t \in [H]$, let $e_t \in \mathcal{E}_t = (\mathcal{A}\mathcal{R}\mathcal{O})^{t+1}$ be an episode prefix in $\mathbf{R}$, $\phi(e_t) \in \mathcal{E}'_t = (\mathcal{A}\mathcal{R}\mathcal{Q})^{t+1}$ be its Markov transformation and $e'_t \in \mathcal{E}'_t$ be an episode of the associated MDP. The function $\phi : \mathcal{E} \to \mathcal{E}'$ transforms the observations according to $\bar{\tau}$, and preserves actions and rewards. The statement says that $\mathbb{P}(\phi(e_t) \mid \mathbf{R}, \pi) = \mathbb{P}(e'_t \mid \mathbf{M_R}, \pi_r)$ (note that $\phi(e_t)$ and $e'_t$ are distinct random variables). We prove this by induction. For $t = 0$, we recall that the irrelevant quantities $a_0, r_0$ are constant and,

$$\mathbb{P}(\phi(a_0 r_0 o_0) = a_0 r_0 q \mid \mathbf{R}, \pi) = \sum_{o \in \mathcal{O}} \mathbb{I}(\tau(q_0, a_0 o) = q)\, \theta_o(q_0, a_0, o)$$
$$= T(q_0, a_0, q)$$
$$= \mathbb{P}(e'_0 = a_0 r_0 q \mid \mathbf{M_R}, \pi_r) \tag{4}$$

where $T : \mathcal{Q} \times \mathcal{A} \to \Delta(\mathcal{Q})$ is the transition function of $\mathbf{M_R}$, from Definition 3. Due to the role of the dummy action, $T(q_0, a_0)$ is the initial distribution of the MDP.

For the inductive step, assume that $\mathbb{P}(\phi(e_{t-1}) \mid \mathbf{R}, \pi) = \mathbb{P}(e'_{t-1} \mid \mathbf{M_R}, \pi_r)$. Then, for any $e' \in \mathcal{E}'_{t-1}$, $arq \in \mathcal{A}\mathcal{R}\mathcal{Q}$, if $q'$ is the last element of $e'$, we have

$$\mathbb{P}(\phi(e_t) = e' arq \mid \mathbf{R}, \pi) = \mathbb{P}(\phi(e_{t-1}) = e' \mid \mathbf{R}, \pi)\, \mathbb{P}(a_t r_t q_{t+1} = arq \mid \phi(e_{t-1}) = e', \mathbf{R}, \pi) \tag{5}$$
$$= \mathbb{P}(e'_{t-1} = e' \mid \mathbf{M_R}, \pi_r)\, \mathbb{P}(a_t r_t q_{t+1} = arq \mid q_t = q', \mathbf{R}, \pi) \tag{6}$$
$$= \mathbb{P}(e'_{t-1} = e' \mid \mathbf{M_R}, \pi_r)\, \pi_r(q', a)\, \theta_r(q', a, r) \sum_{o \in \mathcal{O}} \theta_o(q', a, o)\, \mathbb{I}(q = \tau(q', ao)) \tag{7}$$
$$= \mathbb{P}(e'_{t-1} = e' \mid \mathbf{M_R}, \pi_r)\, \pi_r(q', a)\, \theta_r(q', a, r)\, T(q', a, q) \tag{8}$$
$$= \mathbb{P}(e'_t = e' arq \mid \mathbf{M_R}, \pi_r) \tag{9}$$

where, in (6), we have used the induction hypothesis and the fact that $a_t r_t q_{t+1}$ are Markov in $q'$ by regularity of the policy. $\qquad\square$

Thanks to this relation, the values of corresponding policies are also related in the following way.

**Proposition 4.** *Let $\pi \in \Pi_\mathbf{R}$ be a regular policy in $\mathbf{R}$ such that $\pi(h, a) = \pi_r(\bar{\tau}(h), a)$. Then $\mathbb{E}[V_{0,\mathbf{R}}^\pi] = \mathbb{E}[V_{0,\mathbf{M_R}}^{\pi_r}]$, where $V_{0,\mathbf{R}}^\pi$ and $V_{0,\mathbf{M_R}}^{\pi_r}$ are the values in the respective decision process, and $\mathbb{E}[V_{0,\mathbf{R}}^*] = \mathbb{E}[V_{0,\mathbf{M_R}}^*]$, where expectations are with respect to randomness in $o_0$.*

*Proof.* The statement is composed of two parts. First, we show that $\mathbb{E}[V_{0,\mathbf{R}}^\pi] = \mathbb{E}[V_{0,\mathbf{M_R}}^{\pi_r}]$, which is a direct consequence of Proposition 3. Following the same convention as in the proof of Proposition 3,

we use $\mathcal{E}'_t = (\mathcal{A}\mathcal{R}\mathcal{Q})^{t+1}$ and $\phi$ for the Markov transformation. Then,

$$\mathbb{E}[V_{0,\mathbf{R}}^\pi] = \sum_{r_1 \dots r_H \in \mathcal{R}^{H+1}} \mathbb{P}(r_{1:H} = r_1 \dots r_H \mid \mathbf{R}, \pi) \sum_{i=1}^H r_i \tag{10}$$

$$= \sum_{e' \in \mathcal{E}'_H} \mathbb{P}(\phi(e_H) = e' \mid \mathbf{R}, \pi) \sum_{i=1}^H r_i \tag{11}$$

$$= \sum_{e' \in \mathcal{E}'_H} \mathbb{P}(e'_H = e' \mid \mathbf{M_R}, \pi_{\mathsf{r}}) \sum_{i=1}^H r_i \tag{12}$$

$$= \mathbb{E}[V_{0,\mathbf{M_R}}^{\pi_{\mathsf{r}}}] \tag{13}$$

For the second part of the statement, let $\Pi_\mathbf{R}$ and $\Pi_\mathbf{M}$ be the regular and the Markov policies in $\mathbf{R}$ and $\mathbf{M_R}$, respectively. Then, using Proposition 2 and the first part of this statement,

$$V_{0,\mathbf{R}}^* = \max_{\pi \in \Pi_\mathbf{R}} \mathbb{E}[V_{0,\mathbf{R}}^\pi] = \max_{\pi \in \Pi_\mathbf{R}} \mathbb{E}[V_{0,\mathbf{M_R}}^{\pi_{\mathsf{r}}}] = \max_{\pi_{\mathsf{r}} \in \Pi_\mathbf{M}} \mathbb{E}[V_{0,\mathbf{M_R}}^{\pi_{\mathsf{r}}}] = V_{0,\mathbf{M_R}}^* \tag{14}$$

$\square$

Corollary 5 is a consequence of the two parts of Proposition 4. Since $V_{0,\mathbf{R}}^\pi = V_{0,\mathbf{M_R}}^{\pi_{\mathsf{r}}}$ and the value achieved by the optimal policy in the respective model is the same, $\varepsilon$-optimality of $\pi$ in $\mathbf{R}$ implies $\varepsilon$-optimality of $\pi_{\mathsf{r}}$ in $\mathbf{M_M}$, and vice versa.

## C   Sample Complexity of ADACT–H

In this section we prove high-probability upper bounds on the sample complexity of ADACT–H.

### C.1   Preliminaries

We first state Hoeffding's inequality for Bernoulli variables. In what follows we take $\log$ to be the natural logarithm.

**Lemma 10** (Hoeffding's inequality). *Let $X_1, \dots, X_N$ be $N$ independent random Bernoulli variables with the same expected value $\mathbb{E}[X_1] = p$, and let $\widehat{p}_N = \sum_{i=1}^N X_i / N$ be an empirical estimate of $p$. Then, for any $\delta \in (0,1)$,*

$$\mathbb{P}\left( |\widehat{p}_N - p| \geq \sqrt{\frac{\log(2/\delta)}{2N}} \right) \leq \delta. \tag{15}$$

An alternative to Hoeffding's inequality is the empirical Bernstein inequality, which can be expressed as follows for Bernoulli variables [70, 71].

**Lemma 11** (Empirical Bernstein inequality). *Let $X_1, \dots, X_N$ be $N$ independent random Bernoulli variables with the same expected value $\mathbb{E}[X_1] = p$, and let $\widehat{p}_N = \sum_{i=1}^N X_i / N$ be an empirical estimate of $p$. Then, for any $\delta \in (0,1)$,*

$$\mathbb{P}\left( |\widehat{p}_N - p| \geq \sqrt{\frac{2\widehat{p}\log(4/\delta)}{N}} + \frac{14\log(4/\delta)}{3N} \right) \leq \delta. \tag{16}$$

If $X \sim p_X$ is a discrete random variable, the entropy of $X$ is $H(X) = -\sum_{x \in \mathcal{X}} p_X(x) \log p_X(x)$. Further, for $x \in (0,1)$, we define the binary entropy function as $H_2(x) = -x\log(x) - (1-x)\log(1-x)$. If $(X,Y) \sim p_{XY}$ are two discrete variables, the conditional entropy is $H(Y \mid X) = \sum_{x \in \mathcal{X}} p_X(x) H(Y \mid X = x)$. The mutual information is $I(X;Y) = I(Y;X) = D_{\mathrm{KL}}(p_{XY} \parallel p_X \cdot p_Y)$, where $D_{\mathrm{KL}}$ is the Kullback–Leibler divergence. If $X, Y, Z$ are three random variables, we write $X \to Y \to Z$ if the conditional distribution of $Z$ does not depend on $X$, given $Y$. With these definition, we state Fano's inequality, as one can find in Cover and Thomas [72], (2.140).

**Theorem 12** (Fano's inequality). *Let $X \to Y \to \hat{X}$, for $X, \hat{X} \in \mathcal{X}$ and $P_e = \mathbb{P}(\hat{X} \neq X)$. Then,*

$$H_2(P_e) + P_e \log(|\mathcal{X}| - 1) \geq H(X \mid Y). \tag{17}$$

## C.2 Proof of Theorem 6

In this section we prove Theorem 6, which states a high-probability upper bound on the sample complexity of ADACT–H. The first two lemmas are adaptations of Lemmas 19 and 20 in Balle et al. [10] to the episodic setting.

**Lemma 13.** *For $t \in [H]$, let $\mathcal{X}_1$ and $\mathcal{X}_2$ be multisets sampled from distributions $p_1$ and $p_2$ in $\Delta(\mathcal{E}_{H-t-1})$. If $p_1 = p_2$, then* TESTDISTINCT$(t, \mathcal{X}_1, \mathcal{X}_2, \delta)$ *returns False with probability $1 - \delta$.*

*Proof.* For each $i \in \{1, 2\}$ and each trace $e \in \mathcal{E}_{H-t-1}$, we can view each episode as a random Bernoulli variable with expected value $p_i(e)$ that takes value 1 if we observe $e$, and 0 otherwise. Let $\widehat{p}_i(e) = \sum_{x \in \mathcal{X}_i} \mathbb{I}(x = e)/|\mathcal{X}_i|$ be the empirical estimate of $p_i$, i.e. the proportion of elements in $\mathcal{X}_i$ equal to $e$. For each $i \in \{1, 2\}$, each $u \in [H - t - 1]$ and each prefix $e_{0:u} \in \mathcal{E}_u$, Hoeffding's inequality yields

$$\mathbb{P}\left( |\widehat{p}_i(e_{0:u}*) - p_i(e_{0:u}*)| \geq \sqrt{\frac{\log(2/\delta_s)}{2|\mathcal{X}_i|}} \right) \leq \delta_s.$$

The total number of non-empty prefixes of $\mathcal{E}_{H-t-1}$ equals a geometric sum:

$$(ARO)^1 + \cdots + (ARO)^{H-t} = \frac{(ARO)^{H+1-t} - 1}{ARO - 1} - 1 \leq 2(ARO)^{H-t}.$$

Choosing $\delta_s = \delta/4(ARO)^{H-t}$ and taking a union bound implies that the above inequality holds for each $i \in \{1, 2\}$ and each $e_{0:u}$ simultaneously with probability $1 - 4(ARO)^{H-t}\delta_s = 1 - \delta$, implying

$$L_\infty^p(\mathcal{X}_1, \mathcal{X}_2) = \max_{u, e_{0:u}} |\widehat{p}_1(e_{0:u}*) - \widehat{p}_2(e_{0:u}*)| \leq L_\infty^p(p_1, p_2) + \sqrt{\frac{\log(2/\delta_s)}{2|\mathcal{X}_1|}} + \sqrt{\frac{\log(2/\delta_s)}{2|\mathcal{X}_2|}}$$

$$\leq 0 + 2\sqrt{\frac{\log(2/\delta_s)}{2\min(|\mathcal{X}_1|, |\mathcal{X}_2|)}} = \sqrt{\frac{2\log(8(ARO)^{H-t}/\delta)}{\min(|\mathcal{X}_1|, |\mathcal{X}_2|)}},$$

which is precisely the condition under which TESTDISTINCT$(t, \mathcal{X}_1, \mathcal{X}_2, \delta)$ returns False. $\square$

**Lemma 14.** *For $t \in [H]$, let $\mathcal{X}_1$ and $\mathcal{X}_2$ be multisets sampled from distributions $p_1$ and $p_2$ in $\Delta(\mathcal{E}_{H-t-1})$. If the $L_\infty^p$-distinguishability of $\pi^b$ is $\mu_0$, then* TESTDISTINCT$(t, \mathcal{X}_1, \mathcal{X}_2, \delta)$ *returns True with probability $1 - \delta$ provided that*

$$\min(|\mathcal{X}_1|, |\mathcal{X}_2|) \geq \frac{8}{\mu_0^2} \left( \log(2(ARO)^{H-t}) + \log(4/\delta) \right).$$

*Proof.* Using the same argument as in the proof of Lemma 13, Hoeffding's inequality yields

$$\mathbb{P}\left( |\widehat{p}_i(e_{0:u}*) - p_i(e_{0:u}*)| > \sqrt{\frac{\log(2/\delta_s)}{2|\mathcal{X}_i|}} \right) \leq \delta_s,$$

with the inequality holding simultaneously for $i \in \{1, 2\}$ and each prefix $e_{0:u}$ with probability $1 - \delta$ by choosing $\delta_s = \delta/4(ARO)^{H-t}$. Choosing $\mu_0 \geq 4\sqrt{\log(2/\delta_s)/2|\mathcal{X}_i|}$ for each $i \in \{1, 2\}$ yields

$$|\mathcal{X}_i| \geq \min(|\mathcal{X}_1|, |\mathcal{X}_2|) \geq \frac{8}{\mu_0^2} \log(2/\delta_s) = \frac{8}{\mu_0^2} \left( \log(2(ARO)^{H-t}) + \log(4/\delta) \right).$$

In this case we have

$$L_\infty^p(\mathcal{X}_1, \mathcal{X}_2) = \max_{u, e_{0:u}} |\widehat{p}_1(e) - \widehat{p}_2(e)| \geq L_\infty^p(p_1, p_2) - \sqrt{\frac{\log(2/\delta_s)}{2|\mathcal{X}_1|}} - \sqrt{\frac{\log(2/\delta_s)}{2|\mathcal{X}_2|}}$$

$$\geq \mu_0 - \frac{\mu_0}{4} - \frac{\mu_0}{4} = \frac{\mu_0}{2} \geq 2\sqrt{\frac{\log(2/\delta_s)}{2\min(|\mathcal{X}_1|, |\mathcal{X}_2|)}} = \sqrt{\frac{2\log(8(ARO)^{H-t}/\delta)}{\min(|\mathcal{X}_1|, |\mathcal{X}_2|)}},$$

which is precisely the condition under which TESTDISTINCT$(t, \mathcal{X}_1, \mathcal{X}_2, \delta)$ returns True. $\square$

We are now ready to prove Theorem 6, which we restate below:

**Theorem 6.** *Consider a dataset $\mathcal{D}$ of episodes sampled from an RDP $\mathbf{R}$ and a regular policy $\pi^{\mathsf{b}} \in \Pi_{\mathbf{R}}$. With probability $1 - \delta$, the output of $\text{ADACT–H}(\mathcal{D}, \delta/(2QAO))$ is the transition function of the minimal RDP equivalent to $\mathbf{R}$, provided that $|\mathcal{D}| \geq N_\delta$, where*

$$N_\delta := \frac{21\log(8QAO/\delta)}{d^{\mathsf{b}}_{\min}\mu_0}\sqrt{H\log(2ARO)} \in \tilde{\mathcal{O}}\left(\frac{\sqrt{H}}{d^{\mathsf{b}}_{\min}\mu_0}\right),$$

$d^{\mathsf{b}}_{\min} := \min\{d^{\mathsf{b}}_t(q, ao) \mid t \in [H], q \in \mathcal{Q}_t, ao \in \mathcal{AO}, d^{\mathsf{b}}_t(q, ao) > 0\}$ *is the minimal occupancy distribution, and $\mu_0$ is the $L^{\mathsf{p}}_\infty$-distinguishability.*

*Proof.* The proof consists in choosing $N$ and $\delta$ such that the condition in Lemma 14 is true with high probability for each application of TESTDISTINCT. Consider an iteration $t \in [H]$ of ADACT–H. For a candidate state $qao \in \mathcal{Q}_{\mathsf{c},t+1}$, its associated probability is $d^{\mathsf{b}}_t(q, ao)$ with empirical estimate $\widehat{p}_t(qao) = |\mathcal{X}(qao)|/N$, i.e. the proportion of episodes in $\mathcal{D}$ that are consistent with $qao$. We can apply the empirical Bernstein inequality in (16) to show that

$$\mathbb{P}\left(\left|\widehat{p}_t(qao) - d^{\mathsf{b}}_t(q, ao)\right| \geq \sqrt{\frac{2\widehat{p}_t(qao)\ell}{N}} + \frac{14\ell}{3N} = \frac{\sqrt{2M\ell} + 14\ell/3}{N}\right) \leq \delta,$$

where $M = |\mathcal{X}(qao)|$, $\ell = \log(4/\delta)$, and $\delta$ is the failure probability of ADACT–H. To obtain a bound on $M$ and $N$, assume that we can estimate $d^{\mathsf{b}}_t(q, ao)$ with accuracy $d^{\mathsf{b}}_t(q, ao)/2$, which yields

$$\frac{d^{\mathsf{b}}_t(q, ao)}{2} \geq \frac{\sqrt{2M\ell} + 14\ell/3}{N} \tag{18}$$

$$\widehat{p}_t(qao) \geq d^{\mathsf{b}}_t(q, ao) - \frac{\sqrt{2M\ell} + 14\ell/3}{N} \geq d^{\mathsf{b}}_t(q, ao) - \frac{d^{\mathsf{b}}_t(q, ao)}{2} = \frac{d^{\mathsf{b}}_t(q, ao)}{2}. \tag{19}$$

Combining these two results, we obtain

$$M = N\widehat{p}_t(qao) \geq Nd^{\mathsf{b}}_t(q, ao)/2 \geq \frac{N}{2N}\left(\sqrt{2M\ell} + 14\ell/3\right) = \frac{1}{2}\left(\sqrt{2M\ell} + 14\ell/3\right). \tag{20}$$

Solving for $M$ yields $M \geq 4\ell$, which is subsumed by the bound on $M$ in Lemma 14 since $\mu_0 < 1$. Hence the bound on $M$ in Lemma 14 is sufficient to ensure that we estimate $d^{\mathsf{b}}_t(q, ao)$ with accuracy $d^{\mathsf{b}}_t(q, ao)/2$. We can now insert the bound on $M$ from Lemma 14 into (18) to obtain a bound on $N$:

$$N \geq \frac{2(\sqrt{2M\ell} + 14\ell/3)}{d^{\mathsf{b}}_t(q, ao)} \geq \frac{2\ell}{d^{\mathsf{b}}_t(q, ao)}\left(\frac{4}{\mu_0}\sqrt{\frac{(H-t)\log(2ARO)}{\ell}} + 1 + \frac{14}{3}\right) \equiv N_1. \tag{21}$$

To simplify the bound, we can choose any value larger than $N_1$:

$$N_1 \leq \frac{2\ell}{d^{\mathsf{b}}_t(q, ao)}\left(\frac{4}{\mu_0}\sqrt{H\log(2ARO) + H\log(2ARO)} + \frac{14}{3\mu_0}\sqrt{H\log(2ARO)}\right)$$

$$< \frac{21\ell}{d^{\mathsf{b}}_{\min}\mu_0}\sqrt{H\log(2ARO)} \equiv N_0, \tag{22}$$

where we have used $d^{\mathsf{b}}_t(q, ao) \geq d^{\mathsf{b}}_{\min}$, $\mu_0 < 1$, $\ell = \log 4 + \log(1/\delta) \geq 1$, $H\log(2ARO) \geq \log 4 \geq 1$ and $4\sqrt{2} + 14/3 < \frac{21}{2}$. Choosing $\delta = \delta_0/2QAO$, a union bound implies that accurately estimating $d^{\mathsf{b}}_t(q, ao)$ for each candidate state $qao$ and accurately estimating $p(e_{0:u}*)$ for each prefix in the multiset $\mathcal{X}(qao)$ associated with $qao$ occurs with probability $1 - 2QAO\delta = 1 - \delta_0$, since there are at most $QAO$ candidate states. Substituting the expression for $\delta$ in $N_0$ yields the bound in the theorem.

It remains to show that the resulting RDP is minimal. We show the result by induction. The base case is given by the set $\mathcal{Q}_0$, which is clearly minimal since it only contains the initial state $q_0$. For $t \in [H]$, assume that the algorithm has learned a minimal RDP for sets $\mathcal{Q}_0, \ldots, \mathcal{Q}_t$. Let $\mathcal{Q}_{t+1}$ be the set of states at layer $t + 1$ of a minimal RDP. Due to Proposition 1, each pair of histories that map to a state $q_{t+1} \in \mathcal{Q}_{t+1}$ generate the same probability distribution over suffixes. Hence by Lemma 13, with high probability TESTDISTINCT$(t, \mathcal{X}(qao), \mathcal{X}(q'a'o'), \delta)$ returns false for each pair of candidate states $qao$ and $q'a'o'$ that map to $q_{t+1}$. Consequently, the algorithm

merges $qao$ and $q'a'o'$. On the other hand, by assumption, each pair of histories that map to different states of $\mathcal{Q}_{t+1}$ have $L_\infty^p$-distinguishability $\mu_0$. Hence by Lemma 14, with high probability TESTDISTINCT$(t, \mathcal{X}(qao), \mathcal{X}(q'a'o'), \delta)$ returns true for each pair of candidate states $qao$ and $q'a'o'$ that map to different states in $\mathcal{Q}_{t+1}$. Consequently, the algorithm does not merge $qao$ and $q'a'o'$. It follows that with high probability, ADACT–H will generate exactly the set $\mathcal{Q}_{t+1}$, which is that of a minimal RDP. $\qquad\square$

### C.3 Proof of Theorem 8

In this section we prove Theorem 8, which states an alternative upper bound on the sample complexity of ADACT–H. The proof requires an alternative definition of the algorithm, which we call ADACT–H–A (for "approximation").

**Theorem 8.** *Consider a dataset $\mathcal{D}$ of episodes sampled from an RDP $\mathbf{R}$ and a regular policy $\pi^b \in \Pi_{\mathbf{R}}$. With probability $1 - \delta$, the output of ADACT–H–A, called with $\mathcal{D}$, $\delta/(2QAO)$ and $\varepsilon \in (0, H]$ in input, is the transition function of an $\varepsilon/2$-approximate RDP $\mathbf{R}'$, provided that $|\mathcal{D}| \geq N'_\delta$, where*

$$N'_\delta := \frac{504 H Q A O C^*_{\mathbf{R}'} \log(16 Q A O/\delta)}{\varepsilon \, \mu_0} \sqrt{H \log(2 A R O)} \in \widetilde{\mathcal{O}} \left( \frac{H^{3/2} Q A O C^*_{\mathbf{R}'}}{\varepsilon \, \mu_0} \right).$$

---

**Function** ADACT–H–A$(\mathcal{D}, \delta, \varepsilon, \overline{Q}, \overline{C})$

**Input:** Dataset $\mathcal{D}$, failure probability $0 < \delta < 1$, accuracy $\varepsilon$, upper bounds $\overline{Q}$ on $|\mathcal{Q}'|$ and $\overline{C}$ on $C^*_{\mathbf{R}'}$
**Output:** Set $\mathcal{Q}'$ of RDP states, transition function $\tau' : \mathcal{Q}' \times \mathcal{AO} \to \mathcal{Q}'$ of an approximate RDP $\mathbf{R}'$

1   $\mathcal{Q}'_0 \leftarrow \{q_0\}, \mathcal{X}(q_0) \leftarrow \mathcal{D}$                          // initial state
2   $\mathcal{Q}'_0 \leftarrow \mathcal{Q}'_0 \cup \{q_0^e\}, \mathcal{X}(q_0^e) \leftarrow \emptyset$                  // initial side state
3   **for** $t = 0, \ldots, H$ **do**
4      $\mathcal{Q}'_{t+1} \leftarrow \{q_{t+1}^e\}$                                 // side state
5      **foreach** $ao \in \mathcal{AO}$ **do** $\tau'(q_t^e, ao) = q_{t+1}^e, \mathcal{X}(q_{t+1}^e) \leftarrow \{e_{t+1:H} \mid aroe_{t+1:H} \in \mathcal{X}(q_t^e)\}$
6      $\mathcal{Q}'_{c,t+1} \leftarrow \{qao \mid q \in \mathcal{Q}'_t, ao \in \mathcal{AO}\}$           // get candidate states
7      **foreach** $qao \in \mathcal{Q}'_{c,t+1}$ **do** $\mathcal{X}(qao) \leftarrow \{e_{t+1:H} \mid aroe_{t+1:H} \in \mathcal{X}(q)\}$     // compute suffixes
8      $q_m a_m o_m \leftarrow \arg\max_{qao \in \mathcal{Q}'_{c,t+1}} |\mathcal{X}(qao)|$           // most common candidate
9      $\mathcal{Q}'_{t+1} \leftarrow \mathcal{Q}'_{t+1} \cup \{q_m a_m o_m\}, \tau'(q_m, a_m o_m) = q_m a_m o_m$        // promote candidate
10     $\mathcal{Q}'_{c,t+1} \leftarrow \mathcal{Q}'_{c,t+1} \setminus \{q_m a_m o_m\}$           // remove from candidate states
11     **for** $qao \in \mathcal{Q}'_{c,t+1}$ such that $|\mathcal{X}(qao)|/N \geq \varepsilon/(4\overline{Q}AOH\overline{C})$ **do**
12        $Similar \leftarrow \{q' \in \mathcal{Q}'_{t+1} \mid$ not TESTDISTINCT$(t, \mathcal{X}(qao), \mathcal{X}(q'), \delta)\}$    // confidence test
13        **if** $Similar = \emptyset$ **then** $\mathcal{Q}'_{t+1} \leftarrow \mathcal{Q}'_{t+1} \cup \{qao\}, \tau'(q, ao) = qao$      // promote candidate
14        **else** $q' \leftarrow$ element in $Similar, \tau'(q, ao) = q', \mathcal{X}(q') \leftarrow \mathcal{X}(q') \cup \mathcal{X}(qao)$    // merge states
15        **if** $|\mathcal{Q}'_0| + \cdots + |\mathcal{Q}'_{t+1}| > \overline{Q}$ **then** **return** Failure
16     **end**
17     **for** $qao \in \mathcal{Q}'_{c,t+1}$ such that $|\mathcal{X}(qao)|/N < \varepsilon/(4\overline{Q}AOH\overline{C})$ **do**
18        $\tau'(q, ao) = q_{t+1}^e, \mathcal{X}(q_{t+1}^e) \leftarrow \mathcal{X}(q_{t+1}^e) \cup \mathcal{X}(qao)$          // merge with side state
19     **end**
20 **end**
21 **return** $\mathcal{Q}'_0 \cup \cdots \cup \mathcal{Q}'_{H+1}, \tau'$

---

*Proof.* ADACT–H–A returns the set of RDP states $\mathcal{Q}'$ and transition function $\tau'$ of an approximate RDP $\mathbf{R}'$, taking as input the accuracy $\varepsilon$, an upper bound $\overline{Q}$ on $|\mathcal{Q}'|$, and an upper bound $\overline{C}$ on the concentrability $C^*_{\mathbf{R}'}$ of $\mathbf{R}'$. If, at any moment, the number of RDP states $|\mathcal{Q}'|$ exceeds $\overline{Q}$, the algorithm returns Failure (line 15). ADACT–H–A defines a sequence of side states $q_0^e, \ldots, q_{H+1}^e$ (lines 2 and 4), and defines $\tau'(q_t^e, ao) = q_{t+1}^e$ for each $t \in [H]$ and $ao \in \mathcal{AO}$ (line 5). For each candidate state $qao \in \mathcal{Q}'_{c,t+1}$ such that $|\mathcal{X}(qao)|/N \geq \varepsilon/(4\overline{Q}AOH\overline{C})$, the definition of ADACT–H–A is the same as that of ADACT–H, including the call to TESTDISTINCT (lines 11-14). For each candidate state $qao \in \mathcal{Q}'_{c,t+1}$ such that $|\mathcal{X}(qao)|/N < \varepsilon/(4\overline{Q}AOH\overline{C})$, instead of mapping $(q, ao)$ to the correct RDP state, ADACT–H–A maps $(q, ao)$ to the side state $q_{t+1}^e$ (lines 17-18). Once in $q_{t+1}^e$, $\mathbf{R}'$ remains in a side state for the rest of the episode. The side states do not satisfy Proposition 1, since the

histories that map to side states may assign different probabilities to suffixes (and TESTDISTINCT is never called).

We define an alternative occupancy measure $d'_t(q, ao)$ associated with the approximate RDP $\mathbf{R}'$ and the behavior policy $\pi^\mathsf{b}$. The new definition is given by $d'_0(q_0, a_0 o_0) = \theta_\mathsf{o}(q_0, a_0, o_0)$ and

$$d'_t(q_t, a_t o_t) = \sum_{(q, ao) \in \tau'^{-1}(q_t)} d'_{t-1}(q, ao) \cdot \pi^\mathsf{b}(q_t, a_t) \cdot \theta_\mathsf{o}(q_t, a_t, o_t), \quad t > 0.$$

The only difference between $d'_t$ and $d^\mathsf{b}_t$ is that $d'_t$ is defined with respect to the transition function $\tau'$ of the approximate RDP $\mathbf{R}'$, instead of the transition function $\tau$ associated with the original RDP $\mathbf{R}$. Note that apart from the side states, $\mathbf{R}'$ will contain the same states as $\mathbf{R}$, as long as the candidate states satisfy the condition on line 11, and $\tau'$ will be the same as $\tau$ on those states. Since the states and behavior policy are the same, the $L^\mathsf{p}_\infty$-distingishability $\mu_0$ of $\mathbf{R}'$ will be the same as that of $\mathbf{R}$.

First consider each candidate state $qao \in \mathcal{Q}'_{\mathsf{c},t+1}$ such that $|\mathcal{X}(qao)|/N \geq \varepsilon/(4\overline{Q}AOH\overline{C})$. In this case, ADACT–H–A calls TESTDISTINCT, so Lemmas 13 and 14 apply to these candidate states. The associated occupancy is $d'_t(q, ao)$ with empirical estimate $\widehat{p}_t(qao) = |\mathcal{X}(qao)|/N$. Hence the empirical Bernstein inequality applies to $d'_t(q, ao)$ and $\widehat{p}_t(qao)$. Just as in the proof of Theorem 6, we choose $\mathcal{X}(qao)$ large enough to accurately estimate $d'_t(q, ao)$ within a factor $d'_t(q, ao)/2$ with probability $1 - \delta$. We thus obtain an alternative upper bound on $d'_t(q, ao)$ as follows:

$$d'_t(q, ao) \geq \frac{|\mathcal{X}(qao)|}{N} - \frac{d'_t(q, ao)}{2} \quad \Leftrightarrow \quad \frac{3d'_t(q, ao)}{2} \geq \frac{|\mathcal{X}(qao)|}{N} \geq \frac{\varepsilon}{4\overline{Q}AOH\overline{C}}.$$

From here, we can use the proof of Theorem 6 by substituting $d'_t$ for $d^\mathsf{b}_t$, up until the definition of the bound $N_1$ on $|\mathcal{D}|$ in (21). Inserting the bound on $d'_t(q, ao)$ into the expression for $N_1$ yields

$$N_1 \leq \frac{2\ell}{d'_t(q, ao)} \left( \frac{4}{\mu_0} \sqrt{H \log(2ARO) + H \log(2ARO)} + \frac{14}{3\mu_0} \sqrt{H \log(2ARO)} \right)$$

$$\leq \frac{126\overline{Q}AOH\overline{C}\ell}{\varepsilon\mu_0} \sqrt{H \log(2ARO)} \equiv N_2. \tag{23}$$

Next, consider each candidate state $qao \in \mathcal{Q}'_{\mathsf{c},t+1}$ such that $|\mathcal{X}(qao)|/N < \varepsilon/(4\overline{Q}AOH\overline{C})$. In this case, we instead choose $\mathcal{X}(qao)$ large enough to estimate $d'_t(q, ao)$ with accuracy $\beta$ with probability $1 - \delta$. From the empirical Bernstein inequality, estimating $d'_t(q, ao)$ with accuracy $\beta$ implies

$$\beta \geq \sqrt{\frac{2\widehat{p}_t(qao)\ell}{N}} + \frac{14\ell}{3N} \quad \Leftrightarrow \quad N \geq \frac{2\ell}{\beta} \left( \frac{14}{3} + \frac{\widehat{p}_t(qao)}{\beta} \right) \equiv N_3.$$

Choosing $\beta = \varepsilon/(4\overline{Q}AOH\overline{C})$ implies $\widehat{p}_t(qao) < \beta$, and we can thus simplify $N_3$ as

$$N_3 = \frac{2\ell}{\beta} \left( \frac{14}{3} + \frac{\widehat{p}_t(qao)}{\beta} \right) < \frac{12\ell}{\beta} = \frac{48\overline{Q}AOH\overline{C}\ell}{\varepsilon} \equiv N_4. \tag{24}$$

In addition, this choice of $\beta$ yields the following bound on $d'_t(q, ao)$:

$$d'_t(q, ao) \leq \widehat{p}_t(qao) + \beta < \frac{\varepsilon}{4\overline{Q}AOH\overline{C}} + \frac{\varepsilon}{4\overline{Q}AOH\overline{C}} = \frac{\varepsilon}{2\overline{Q}AOH\overline{C}}.$$

We prove that $\mathbf{R}'$ is an $\varepsilon/2$-approximation of the original RDP $\mathbf{R}$. We briefly overload notation by letting $d^*_t(q, ao)$ refer to the occupancy of an optimal policy $\pi^*$ *with respect to the transition function* $\tau'$ *of* $\mathbf{R}'$. Consider a candidate state $qao \in \mathcal{Q}'_{\mathsf{c},t+1}$ such that $|\mathcal{X}(qao)|/N < \varepsilon/(4\overline{Q}AOH\overline{C})$. The contribution to the expected optimal reward of $\mathbf{R}$ of all histories that map to $qao$ is bounded as

$$d^*_t(q, ao)(H - t) \leq C^*_{\mathbf{R}'} d'_t(q, ao)H < \frac{\varepsilon}{2\overline{Q}AO},$$

since $(H - t)$ is the maximum reward obtained during the remaining time steps. Since $qao$ is mapped to a side state of $\mathbf{R}'$, an optimal policy for $\mathbf{R}'$ may not accurately estimate the expected optimal value for $qao$, but the contribution of all such candidate states to the expected optimal value is at most

$$\sum_{t \in [H-1]} \sum_{q \in \mathcal{Q}_t} \sum_{ao \in \mathcal{AO}} d^*_t(q, ao)(H - t) \leq \sum_{t \in [H-1]} \sum_{q \in \mathcal{Q}_t} \sum_{ao \in \mathcal{AO}} \frac{\varepsilon}{2\overline{Q}AO} \leq \frac{\varepsilon}{2},$$

since there can be at most $\overline{Q}AO$ such candidate states. Hence any optimal policy for $\mathbf{R}'$ is an $\varepsilon/2$-optimal policy for $\mathbf{R}$, which implies that we can approximate an $\varepsilon$-optimal regular policy for the exact RDP $\mathbf{R}$ by finding an $\varepsilon/2$-optimal policy for the approximate RDP $\mathbf{R}'$.

It is easy to verify that the bound $N_4$ in (24) is less than the bound $N_2$ in (23). Hence a worst-case bound is obtained by assuming that $|\mathcal{X}(qao)|/N \geq \varepsilon/(4\overline{Q}AOH\overline{C})$ for each $t \in [H]$ and each candidate state $qao \in \mathcal{Q}'_{c,t+1}$, which yields an upper bound $N_2$. Note that ADACT–H–A takes as input an upper bound $\overline{Q}$ on the number of RDP states $|\mathcal{Q}'|$ of $\mathbf{R}'$, as well as an upper bound $\overline{C}$ of the concentrability coefficient $C^*_{\mathbf{R}'}$. If the learning agent has no prior knowledge of $\overline{Q}$ and $\overline{C}$, it could start with small estimates of $\overline{Q}$ and $\overline{C}$, and in the case that ADACT–H–A returns Failure or the resulting policy has larger concentrability than $\overline{C}$ for $\mathbf{R}'$, iteratively double the estimates $\overline{Q}$ and/or $\overline{C}$ and call the algorithm again. This only increases the computational complexity of ADACT–H–A by a factor $O(\log QC^*_{\mathbf{R}'})$, and the resulting upper bounds $\overline{Q}$ and $\overline{C}$ do not exceed $2Q$ and $2C^*_{\mathbf{R}'}$. Since we already have an estimate $\overline{Q}$, in each iteration we can call ADACT–H–A with $\delta = \delta_1/(2\overline{Q}AO)$ to ensure that the bound $N_2$ holds for each candidate state simultaneously with probability $1 - \delta_1$. Substituting this value of $\delta$ in the bound $N_2$ in (23) and using $\overline{Q} < 2Q$ and $\overline{C} < 2C^*_{\mathbf{R}'}$ yields the sample complexity bound stated in the theorem. $\qquad\square$

**Lemma 15.** *The concentrability $C^*_{\mathbf{R}'}$ of the approximate RDP $\mathbf{R}'$ satisfies*

$$C^*_{\mathbf{R}'} \leq C^*_{\mathbf{R}}(1 + 3\overline{Q}AO).$$

*Proof.* For each $t > 0$, let $d'_t(q^e_t)$ be the occupancy of the side state $q^e_t$ in the approximate RDP $\mathbf{R}'$. We prove by induction on $t$ that $d'_t(q^e_t)$ satisfies

$$d'_t(q^e_t) < \frac{\varepsilon \sum_{u=0}^{t-1} |\mathcal{Q}_u|}{2\overline{Q}H\overline{C}} \leq \frac{\varepsilon}{2H\overline{C}}.$$

The base case is given by $t = 1$. In this case, a candidate state $(q_0, ao)$ is mapped to $q^e_1$ if $d'_t(q_0, ao) = d'_t(q_0, ao) < \varepsilon/(2\overline{Q}AOH\overline{C})$. Since there can be at most $AO = |\mathcal{Q}_0|AO$ such candidate states, we have

$$d'_t(q^e_t) < \frac{\varepsilon|\mathcal{Q}_0|AO}{2\overline{Q}AOH\overline{C}} = \frac{\varepsilon|\mathcal{Q}_0|}{2\overline{Q}H\overline{C}}.$$

For $t > 1$, a candidate state $(q_{t-1}, ao)$ is mapped to $q^e_t$ if $d'_t(q_{t-1}, ao) < \varepsilon/(2\overline{Q}AOH\overline{C})$. Again, there can be at most $|Q_{t-1}|AO$ such candidate states. Since all occupancy of $q^e_{t-1}$ is also mapped to $q^e_t$, we have

$$d'_t(q^e_t) < d'_{t-1}(q^e_{t-1}) + \frac{\varepsilon|\mathcal{Q}_{t-1}|AO}{2\overline{Q}AOH\overline{C}} < \frac{\varepsilon \sum_{u=0}^{t-2} |\mathcal{Q}_u|}{2\overline{Q}H\overline{C}} + \frac{\varepsilon|\mathcal{Q}_{t-1}|}{2\overline{Q}H\overline{C}} = \frac{\varepsilon \sum_{u=0}^{t-1} |\mathcal{Q}_u|}{2\overline{Q}H\overline{C}},$$

where we have used the induction hypothesis.

Consider a candidate state $(q, ao)$ of $\mathbf{R}$ at time $t$. Due to approximation, some histories in $\bar{\tau}^{-1}(q)$ are mapped to side states in $\mathbf{R}'$ instead of $q$, and we can therefore write $d^b_t(q, ao) = d'_t(q, ao) + \xi \leq d'_t(q, ao) + d'_t(q^e_t)$, where $\xi$ is the total occupancy of histories in $\bar{\tau}^{-1}(q)$ mapped to side states. In turn, this implies

$$d^*_t(q, ao) \leq d^b_t(q, ao)C^*_{\mathbf{R}} \leq (d'_t(q, ao) + d'_t(q^e_t))C^*_{\mathbf{R}} < (d'_t(q, ao) + \frac{\varepsilon}{2H\overline{C}})C^*_{\mathbf{R}}.$$

The concentrability of a candidate state $(q, ao)$ in the approximate RDP $\mathbf{R}'$ that is not mapped to a side state (i.e. $d'_t(q, ao) \geq \varepsilon/(6\overline{Q}AOH\overline{C})$) can now be bounded as

$$\frac{d^*_t(q, ao)}{d'_t(q, ao)} < \frac{d'_t(q, ao) + \varepsilon/(2H\overline{C})}{d'_t(q, ao)}C^*_{\mathbf{R}} = \left(1 + \frac{\varepsilon}{2H\overline{C}d'_t(q, ao)}\right)C^*_{\mathbf{R}} \leq C^*_{\mathbf{R}}(1 + 3\overline{Q}AO).$$

This concludes the proof of the lemma. $\qquad\square$

## C.4 Distinguishability Parameters

As defined in the main text, for $t \in [H]$, we consider a metric $L$ over distributions on the remaining part of the episode $\Delta(\mathcal{E}_\ell)$, for $\ell = H - t$. Then, the $L$-distinguishability of an RDP $\mathbf{R}$ and a policy $\pi$ is the maximum $\mu_0$ such that, for any $t \in [H]$ and any two distinct $q, q' \in \mathcal{Q}_t$, the probability distributions over suffix traces $e_{t:H} \in \mathcal{E}_\ell$ from the two states satisfy

$$L(\mathbb{P}(e_{t:H} \mid q_t = q, \pi), \mathbb{P}(e_{t:H} \mid q_t = q', \pi)) \geq \mu_0 \tag{25}$$

So, $\mu_0$ is a feature of the RDP and the policy combined and it quantifies the distance between any two distinct states of the RDP with respect to the distributions they induce over the observable quantities. Distinguishability parameters have been first introduced in Ron et al. [56] and later generalized for other metrics. They can be also found in Balle [59] for PDFA learning and in Ronca and De Giacomo [26], Ronca et al. [27] for RDP learning.

According to the definition we adopt, there exists an $L$-distinguishability for any RDP and policy. However, as stated in Assumption 2, we require $\mu_0$ to be strictly positive. This does not constitute a restriction for the RDP, since it can be always minimized while preserving all conditional probabilities. Though it implies that, in any state, the behavior policy takes with positive probability all actions that are needed to observe episode suffixes that have different probability under the two states. Clearly if this was not the case for two distinct $q, q' \in \mathcal{Q}_t$ at some $t \in [H]$, $\mathbb{P}(e_{t:H} \mid q_t = q, \pi) = \mathbb{P}(e_{t:H} \mid q_t = q', \pi)$ and no information would be available for the algorithm to distinguish $q$ and $q'$. Assumption 2 is implied by Assumption 1. However, it becomes necessary for Theorem 8, since this does not rely on Assumption 1.

The metric selected also influences the actual value of the distinguishability parameter. In this paper, we adopt $L_\infty^{\mathsf{p}}$, as it can be seen from the TESTDISTINCT function in the two algorithms. A more standard distance would be $L_\infty$. According to Eq. (25), an $L_\infty$-distinguishability of $\mu_0$ implies that for any $t \in [H]$ and two distinct $q, q' \in \mathcal{Q}_t$,

$$\max_{e \in \mathcal{E}_{H-t}} |\mathbb{P}(e_{t:H} = e \mid q_t = q) - \mathbb{P}(e_{t:H} = e \mid q_t = q')| \geq \mu_0. \tag{26}$$

This means that some sequence until the end of the episode has a different probability of being generated from the two states. Although similar, the $L_\infty^{\mathsf{p}}$ distance, maximizes for the full trace as well as any of its prefixes as

$$\max_{u \in [H-t], e \in \mathcal{E}_u} |\mathbb{P}(e_{t:H} = e* \mid q_t = q) - \mathbb{P}(e_{t:H} = e* \mid q_t = q')| \geq \mu_0 \tag{27}$$

As it has been discussed in Balle [59], Appendix A.5, the prefix $L_\infty^{\mathsf{p}}$ metric always upper bounds the $L_\infty$ metric, up to a multiplicative factor, while there are pairs of distributions in which $L_\infty$ is exponentially smaller than $L_\infty^{\mathsf{p}}$ with respect to the expected suffix length. This motivates our choice. Moreover, in the specific case of our fixed horizon setting, we have that the $L_\infty^{\mathsf{p}}$-distinguishability is never lower than $L_\infty$-distinguishability. Note that in the hard instance of Appendix E.2, the two coincide.

The lower bound is stated in terms of the $L_1^{\mathsf{p}}$-distinguishability of the RDP. While $L_\infty^{\mathsf{p}}$ is achieved for one specific trace prefix maximizing the difference in probability, $L_1^{\mathsf{p}}$ takes all traces into account as $\sum_{u \in [H-t], e \in \mathcal{E}_u} |\mathbb{P}(e_{t:H} = e* \mid q_t = q) - \mathbb{P}(e_{t:H} = e* \mid q_t = q')|$. Due to this relation, the $L_\infty^{\mathsf{p}}$-distinguishability always lower bounds the $L_1^{\mathsf{p}}$-distinguishability in the fixed horizon setting.

## D  RegORL with Subsampled VI-LCB

In this section we demonstrate the composition of our proposed algorithm with a specific Offline Reinforcement Learning algorithm for MDPs. Specifically, we adopt Subsampled VI-LCB, from Algorithm 3 of Li et al. [16] and report the combined sample complexity of this choice, through a simple application of Theorem 7.

First, we introduce the occupancy distribution and the single-policy conentrability coefficient for MDPs. Let $\mathbf{M} = \langle \mathcal{Q}, \mathcal{A}, \mathcal{R}, T, R, H \rangle$ be an MDP with states $\mathcal{Q}$, horizon $H$, transition function $T : \mathcal{Q} \times \mathcal{A} \to \mathcal{Q}$ and reward function $R : \mathcal{Q} \times \mathcal{A} \to \Delta(\mathcal{R})$. The state-action occupancy distribution of a policy $\pi : \mathcal{Q} \to \Delta(\mathcal{A})$ in $\mathbf{M}$ at step $t \in [H]$ is $d_{\mathbf{m},t}^\pi(q, a) = \mathbb{P}(q_t = q, a_t = a \mid \mathbf{M}, \pi)$. For our

purposes, it suffices to consider a fixed initial state $q_0$. Finally, the MDP single-policy concentrability of a policy $\pi^{\mathsf{b}}$ is [15]:

$$C^* = \max_{t \in [H], q \in \mathcal{Q}, a \in \mathcal{A}} \frac{d_{\mathsf{m},t}^{\pi^*}(q,a)}{d_{\mathsf{m},t}^{\pi^{\mathsf{b}}}(q,a)} \tag{28}$$

We can now express the sample complexity of Subsampled VI-LCB.

**Theorem 16** (Li et al. [16])**.** *Let $\mathcal{D}$ be a dataset of $N_{\mathsf{m}}$ episodes, sampled from an MDP $\mathbf{M}$ with a Markov policy $\pi^{\mathsf{b}}$. For any $\varepsilon \in (0, H]$ and $0 < \delta < 1/12$, with probability exceeding $1 - \delta$, the policy $\widehat{\pi}$ returned by Subsampled VI-LCB obeys $V_0^*(q_0) - V_0^{\widehat{\pi}}(q_0) \leq \varepsilon$, as long as:*

$$N_{\mathsf{m}} \geq \frac{c\, H^3 Q C^* \log \frac{N_{\mathsf{m}} H}{\delta}}{\varepsilon^2} \tag{29}$$

*for a positive constant c.*

The analysis in Li et al. [16] of Subsampled VI-LCB assumes that the reward function is deterministic and known. Thus, restricting our attention to this setting, we consider any episodic RDP with history-dependent, deterministic rewards. The reward function can be regarded as known, since it may be easily extracted from the dataset resulting from the Markov transformation of Definition 2.

**Corollary 17.** *Let $\mathcal{D}$ be a dataset of $N$ episodes, sampled with a regular policy $\pi^{\mathsf{b}} \in \Pi_{\mathbf{R}}$ from an RDP $\mathbf{R}$ with deterministic rewards. If Subsampled VI-LCB is the OFFLINERL algorithm in Algorithm 1, then, for any $\varepsilon \in (0, H]$ and $0 < \delta < 1/12$, with probability exceeding $1 - \delta$, the output of RegORL$(\mathcal{D}, \varepsilon, \delta)$ is an $\varepsilon$-optimal policy of $\mathbf{R}$, as long as*

$$N \geq 2 \max\left\{ \frac{21 \log(8QAO/\delta)}{d_{\min}^{\mathsf{b}} \mu_0} \sqrt{H \log(2ARO)},\ \frac{c\, H^3 Q C_{\mathbf{R}}^* \log \frac{2NH}{\delta}}{\varepsilon^2} \right\} \tag{30}$$

*Proof.* This corollary follows as a direct application of Theorem 16 to Theorem 6. It only remains to verify that the single-policy concentrability of the MDP underlying the dataset $\mathcal{D}'$ that Subsampled VI-LCB receives is $C_{\mathbf{R}}^*$. The dataset $\mathcal{D}'$ is generated according to the Markov transformation $\bar{\tau}$ from Definition 2. We only consider the cases in which ADACT–H succeeds. Let $\pi \in \Pi_{\mathbf{R}}$ be any regular policy and $q_t, q_t'$ the states reached at step $t$ by $\mathbf{R}$ and $\mathbf{M}_{\mathbf{R}}$, respectively. Then for $t > 0$,

$$d_t^{\pi}(q,a) \coloneqq \mathbb{P}(q_t = q \mid \mathbf{R}, \pi)\, \pi_{\mathsf{r}}(q,a) \tag{31}$$
$$= \mathbb{P}(\bar{\tau}(h_{t-1}) = q \mid \mathbf{R}, \pi)\, \pi_{\mathsf{r}}(q,a) \tag{32}$$
$$= \mathbb{P}(q_t' = q \mid \mathbf{M}_{\mathbf{R}}, \pi_{\mathsf{r}})\, \pi_{\mathsf{r}}(q,a) \tag{33}$$
$$= d_{\mathsf{m},t}^{\pi_{\mathsf{r}}}(q,a) \tag{34}$$

This is valid for any regular policy, and for the optimal and behavior policies in particular. Then,

$$C_{\mathbf{R}}^* = \max_{t \in [H], q \in \mathcal{Q}_t, ao \in \mathcal{AO}} \frac{d_t^*(q, ao)}{d_t^{\mathsf{b}}(q, ao)} \tag{35}$$

$$= \max_{t \in [H], q \in \mathcal{Q}_t, ao \in \mathcal{AO}} \frac{d_t^{\pi^*}(q,a)\, \theta_{\mathsf{o}}(q,a,o)}{d_t^{\pi^{\mathsf{b}}}(q,a)\, \theta_{\mathsf{o}}(q,a,o)} \tag{36}$$

$$= \max_{t \in [H], q \in \mathcal{Q}, a \in \mathcal{A}} \frac{d_{\mathsf{m},t}^{\pi_{\mathsf{r}}^*}(q,a)}{d_{\mathsf{m},t}^{\pi_{\mathsf{r}}^{\mathsf{b}}}(q,a)} \tag{37}$$

$$= C^* \tag{38}$$

$\square$

Similarly to the previous corollary, it is also possible to combine Theorem 16 with Theorem 8. In this case, the sample complexity of Subsampled VI-LCB for learning an $\varepsilon/2$-accurate policy with probability $1 - \delta/2$ would be combined with $N'_{\delta/2}$ of Theorem 8.

# E  Sample Complexity Lower Bound: Proof of Theorem 9

In this section, we prove the sample complexity lower bound in Theorem 9. The proof is based on a suitable composition of a two-armed bandit and an instance of the learning problem associated to noisy parity functions. We first describe this latter problem and its sample-complexity lower bound in Appendix E.1. Then, we compose a hard class of RDP instances in Appendix E.2, and prove the final statement in Appendix E.3.

## E.1  Learning parity with noise

Let $\mathbb{B} = \{0, 1\}$ and $L \in \mathbb{N}$. For any string $x \in \mathbb{B}^L$, the parity function $f_x : \mathbb{B}^L \to \mathbb{B}$ is $f_x(y) = \oplus_{i \in [L-1]} x_i y_i$, where $\oplus$ is addition modulo 2. For noise parameter $\xi \in (0, 0.5)$, a noisy parity function $f_{x,\xi}$ returns $f_x(y)$ with probability $0.5 + \xi$ and $1 - f_x(y)$ otherwise. Consider the class of parity functions $\mathbb{F}(L) = \{f_x\}_{x \in \mathbb{B}^L}$ and the class of noisy parity functions $\mathbb{F}(L, \xi) = \{f_{x,\xi}\}_{x \in \mathbb{B}^L}$. Assume that $x, y_1, y_2, \ldots \sim \mathrm{unif}(\mathbb{B}^L)$ are uniformly sampled. The success probability of a streaming algorithm $\mathfrak{A}$ for $\mathbb{F}(L, \xi)$ is the probability that $\mathfrak{A}$ recovers $x$, given in input a sequence of observations $(y_i, f_{x,\xi}(y_i))_i$.

**Lemma 18.** *Any streaming algorithm for $\mathbb{F}(L, \xi)$ with a success probability higher than $O(2^{-L})$ requires at least $\Omega(L/\xi)$ or $2^{\Omega(L)}$ input samples $(y_i, f_{x,\xi}(y_i))_i$.*

*Proof.* Learning in $\mathbb{F}(L, \xi)$ is the problem of recovering $x \in 2^{\mathbb{B}}$ from noisy data $(y_i, b_i)$, where $b_i = f_x(y_i)$ with probability $0.5 + \xi$, and $b_i = 1 - f_x(y_i)$ otherwise. This is the problem of learning in $\mathbb{F}(L)$ with corruption rate $0.5 - \xi$. Hence, we focus on the problem of learning noiseless parity first.

The Statistical Query dimension $\mathrm{SQDIM}(\mathcal{C}, d)$, characterizes the complexity of learning in the class $\mathcal{C}$ with respect to the prior distribution $d \in \Delta(\mathcal{C})$. As defined in [73], $\mathrm{SQDIM}(\mathcal{C}, d)$ is the maximum $n \in \mathbb{N}$ such that there exist distinct $f_1, \ldots, f_n \in \mathcal{C}$, such that their pairwise correlations with respect to $d$ are between $-1/n$ and $1/n$. For the class of parity functions, under the uniform distribution over $x$, $\mathrm{SQDIM}(\mathbb{F}(L), \mathrm{unif}) = 2^L$. This was already observed in [74], for a slightly different notion of SQ dimension. However, to verify this, we can consider a natural ordering over binary strings in $\mathcal{X}$, and represent the problem of learning $\mathbb{F}(L)$ as a matrix $M = (m_{ij}) \in \{1, -1\}^{2^L \times 2^L}$, defined as $m_{ij} = (-1)^{f_{x_j}(y_i)} = (-1)^{y_i \cdot x_j}$, where scalar product is modulo 2. We have that $M$ is a Hadamard matrix. Then, since every row is orthogonal to the others, and the same is true for columns, every couple of parity functions are uncorrelated under the uniform distribution over $x$.

Regarding the noisy parity problem, since $\mathrm{SQDIM}(\mathbb{F}(L), \mathrm{unif}) = 2^L$, we can apply Corollary 8 of [75] with $m = 2^L$, to have that the matrix $M$ corresponding to the parity problem is a $(k, l)$-$L_2$-extractor with error $2^{-r}$, for $k, l, r \in \Omega(L)$. Since $M$ is a suitable extractor, we can apply Theorem 1 of [76], which considers the problem of learning $M$ with the additional noise parameter $\xi$. We obtain that, in the streaming setting, any branching program $B$ for $\mathbb{F}(L, \xi)$ whose depth is at most $2^{f_1(k,l,r)}$ and width is at most $2^{ckl/\xi}$ has a success probability of at most $O(2^{-f_1(k,l,r)})$, where $c$ is a suitable constant and $f_1$ is equation (1) of [76].

Then, if the success probability is not in $O(2^{-f_1(k,l,r)})$, meaning it is higher, we have that the depth of $\mathfrak{A}$ exceeds $2^{f_1(k,l,r)}$ or the width of $\mathfrak{A}$ exceeds $2^{ckl/\xi}$. Since $k, l, r \in \Omega(L)$, then, if the success probability is not in $O(2^{-L})$, the depth of $\mathfrak{A}$ is $2^{\Omega(L)}$ or the width of $\mathfrak{A}$ is $2^{\Omega(L^2)/\xi}$. Width and depth refer to the computational model that represents $\mathfrak{A}$ as a branching program. A branching program is a directed acyclic graph in which internal nodes have one outgoing edge for each possible input sample, that is $|\mathbb{B}^L \times \mathbb{B}| = 2^{L+1}$ in our problem, and leaves correspond to algorithm decisions. From the required width and depth we know that $\mathfrak{A}$ has a leaf in layer $2^{\Omega(L)}$ or in a layer that contains $2^{\Omega(L^2)/\xi}$ nodes. The former case implies a worst case sample complexity requirement that is exponential in $L$. For the latter, we observe that in order to reach that width, at least $\log_{2^{L+1}} 2^{\Omega(L^2/\xi)}$ transitions and input samples, are required. This is $\Omega(L/\xi)$. $\square$

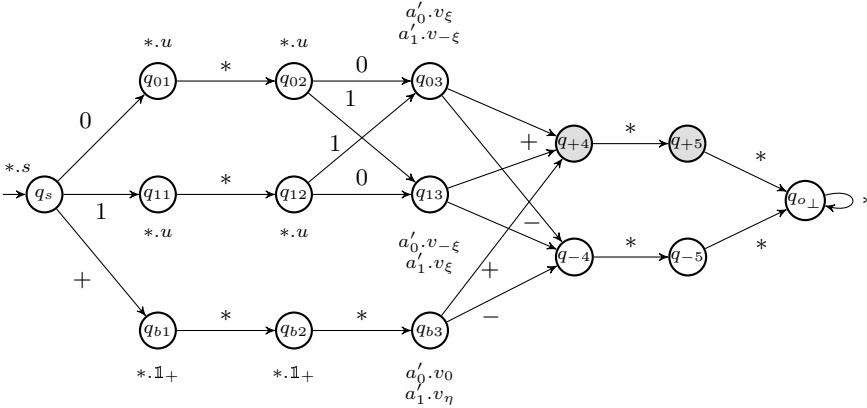

Figure 3: One episodic RDP instance $\mathbf{R}_{101,1} \in \mathbb{R}(3,5,\xi,\eta)$, associated to the parity function $f_{101}$ and optimal arm $a_1'$. The transition function is represented by the arcs, labelled by observations (transitions do not depend on actions). The star denotes any symbol. If the label of a state $q$ is $a.d$, then the observation function is $\theta_{\mathsf{o}}(q,a) = d$, where $d \in \Delta(\mathcal{O})$ (some irrelevant outputs are omitted).

## E.2 Class of hard RDP instances

For our main lower bound, we define a class of hard RDP instances. Figure 3 shows one possible instance in this class. We will soon define it formally, but we can observe that its structure is organized in two main paths. The two branches in the top part encode a parity computation according to some hidden code $x \in \mathbb{B}^L$, so that behaving optimally in that region requires to solve a parity problem (the one of Lemma 18). The bottom part reaches a two-armed bandit whose optimal action is $c$. The right-most states are winning or losing states that provide a positive and null reward accordingly.

Formally, we define a class of hard RDP instances as $\mathbb{R}(L,H,\xi,\eta) = \{\mathbf{R}_{x,c}\}_{x \in \mathbb{B}^L, c \in \{0,1\}}$ where $\mathbf{R}_{x,c} = \langle \mathcal{Q}, \mathcal{A}\mathcal{O}, \Omega, \tau, \theta, q_s, H \rangle$, for $\mathcal{Q} = \{q_s, q_{o_\perp}\} \cup \{q_{0i}, q_{1i}, q_{bi}\}_{i=1,\ldots,L} \cup \{q_{+,i}, q_{-,i}\}_{i=L+1,\ldots,H}$, $\mathcal{A} = \{a_0', a_1'\}$, $\mathcal{O} = \{0, 1, +, -\}$. Assume $L \geq 1$ and $H > L$. Rewards are zero everywhere, except in the winning states

$$\theta_{\mathsf{r}}(q,a,r) = \mathbb{1}_1 \text{ if } q = q_{+i} \text{ with } i > L, \mathbb{1}_0, \text{ otherwise} \tag{39}$$

where we recall that $\mathbb{1}_x$ represents the deterministic distribution on $x$. For observation probabilities, we denote the distributions $u(o) := \text{unif}\{0,1\}$ and

$$v_\alpha(o) := \begin{cases} \frac{1+\alpha}{2} & \text{if } o = + \\ \frac{1-\alpha}{2} & \text{if } o = - \\ 0 & \text{otherwise} \end{cases} \qquad s(o) := \begin{cases} 1/4 & \text{if } o = 0 \\ 1/4 & \text{if } o = 1 \\ 1/2 & \text{if } o = + \end{cases} \tag{40}$$

Now define observations as

$$\theta_{\mathsf{o}}(q,a,o) = \begin{cases} s(o) & \text{if } q = q_s \\ u(o) & \text{if } q \in \{q_{0i}, q_{1i}\}_{i=1,\ldots,L-1} \\ v_\xi(o) & \text{if } q = q_{0L} \wedge a = a_0' \text{ or } q = q_{1L} \wedge a = a_1' \\ v_{-\xi}(o) & \text{if } q = q_{0L} \wedge a = a_1' \text{ or } q = q_{0L} \wedge a = a_1' \\ \mathbb{1}_+(o) & \text{if } q = q_{bi} \text{ with } i < L \\ v_0(o) & \text{if } q = q_{bL} \wedge a = a_0' \\ v_\eta(o) & \text{if } q = q_{bL} \wedge a = a_1' \wedge c = 1 \\ v_{-\eta}(o) & \text{if } q = q_{bL} \wedge a = a_1' \wedge c = 0 \\ \mathbb{1}_{o_\perp}(o) & \text{if } q = q_{o_\perp} \end{cases} \tag{41}$$

Finally, the transition function is defined such that $\bar{\tau}(q_s, h_{L-1}) = q_{iL}$ with $i = f_x(o_{0:L-1})$, and

$$\tau(q_{kL}, a+) = q_{+,L+1} \qquad \tau(q_{kL}, a-) = q_{-,L+1} \qquad \text{for } k = 1, 2 \tag{42}$$

$$\tau(q_{bL}, a+) = q_{+,L+1} \qquad \tau(q_{bL}, a-) = q_{-,L+1} \tag{43}$$

$$\tau(q_{+i}, ao) = q_{+,i+1} \qquad\qquad \tau(q_{-i}, ao) = q_{-,i+1} \tag{44}$$

$$\tau(q_s, a+) = q_{b1} \qquad\qquad \tau(q_{bi}, ao) = q_{b,i+1} \qquad\qquad \text{for } i < L \tag{45}$$

$$\tau(q_{+H}, ao) = q_{o_\perp} \qquad\qquad \tau(q_{-H}, ao) = q_{o_\perp} \qquad\qquad \tau(q_{o_\perp}, a, o) = q_{o_\perp} \tag{46}$$

In addition, the trnasitions $\tau(q_{0i}, ao)$ and $\tau(q_{1i}, ao)$, for $i < L$, are defined according to $o \in \{0, 1\}$ and the parity code $x$. Namely, $\tau(q_{0i}, ao)$ equals $q_{1,i+1}$ iff $o \oplus x(i) = 1$, and $q_{0,i+1}$, otherwise. $\tau(q_{1i}, ao)$ is defined analogously.

### E.3 Proof of Theorem 9

**Theorem 9.** *For any $(C_{\mathbf{R}}^*, H, \varepsilon, \mu_0)$ satisfying $C_{\mathbf{R}}^* \geq 2$, $H \geq 2$ and $\varepsilon \leq H\mu_0/64$, there exists an RDP with horizon $H$, $L_1^{\mathsf{p}}$-distinguishability $\mu_0$ and a regular behavior policy $\pi^{\mathsf{b}}$ with RDP single-policy concentrability $C_{\mathbf{R}}^*$, such that if $\mathcal{D}$ has been generated using $\pi^{\mathsf{b}}$ and $\mathbf{R}$, and*

$$|\mathcal{D}| \notin \Omega\left(\frac{H}{\mu_0} + \frac{C_{\mathbf{R}}^* H^2}{\varepsilon^2}\right) \tag{3}$$

*then, for any algorithm $\mathfrak{A}: \mathcal{D} \mapsto \widehat{\pi}$ returning non-Markov deterministic policies, the probability that $\widehat{\pi}$ is not $\varepsilon$-optimal is at least $1/4$.*

*Proof.* Denote with $\pi^{\mathsf{b}}$ a regular policy in $\mathbf{R}$ and $\mathcal{D} \in \mathbb{D}$ a dataset of episodes of length $H$, collected from $\mathbf{R}$ and the behavior policy $\pi^{\mathsf{b}}$. For an RDP $\mathbf{R}$, let $\Pi_{\mathsf{d}} = \mathcal{A}^{\mathcal{H}}$ be the set of deterministic non-Markov policies and $\mathfrak{A} = \mathbb{D} \to \Pi_{\mathsf{d}}$ an offline RL algorithm. For $\delta < 0.5$, we say that an algorithm $\mathfrak{A}$ is $(\varepsilon, \delta)$-PAC for the class of RDPs $\mathbb{R}$ under $\varphi$, if, for every $\mathbf{R} \in \mathbb{R}$ and $\mathcal{D} \in \mathbb{D}$, if the condition $\varphi(\mathcal{D}, \pi^{\mathsf{b}})$ is verified, then the output policy $\mathfrak{A}(\mathcal{D})$ is $\varepsilon$-optimal in $\mathbf{R}$, with probability $1 - \delta$. One notable case is that of $\varphi$ requiring a minimum dataset size.

Since the output of a generic algorithm might be any generic non-Markov deterministic policy, we cannot restrict our attention to regular policies. We expand the value of a policy $\pi: \mathcal{H} \to \mathcal{A}$ in a RDP $\mathbf{R}_{x,c} \in \mathbb{R}(L, H, \xi, \eta)$ as follows:

$$\mathbb{E}[V_0^\pi(h_0)] = \mathbb{E}\left[\sum_{i=1}^H r_i \mid \pi\right] \tag{47}$$

$$= (H - L)\,\mathbb{P}(q_{L+1} = q_{+,L+1} \mid \pi) \tag{48}$$

$$= (H - L)\sum_{q \in \mathcal{Q}_L} \mathbb{P}(q_L = q \mid \pi)\,\mathbb{P}(q_{L+1} = q_{+,L+1} \mid q_L = q, \pi) \tag{49}$$

$$\begin{aligned}
= {}&(H - L)\,(\mathbb{P}(q_L = q_{0L} \mid \pi)\,\mathbb{P}(q_{L+1} = q_{+,L+1} \mid q_L = q_{0L}, \pi) \\
&+ \mathbb{P}(q_L = q_{1L} \mid \pi)\,\mathbb{P}(q_{L+1} = q_{+,L+1} \mid q_L = q_{1L}, \pi)) \\
&+ (H - L)\,(\mathbb{P}(q_L = q_{bL} \mid \pi)\,\mathbb{P}(q_{L+1} = q_{+,L+1} \mid q_L = q_{bL}, \pi))
\end{aligned} \tag{50}$$

$$\begin{aligned}
= {}&\frac{H - L}{2}\,(\mathbb{P}(q_L = q_{0L} \mid o_0 \in \{0,1\}, \pi)\,\mathbb{P}(q_{L+1} = q_{+,L+1} \mid q_L = q_{0L}, \pi) \\
&+ \mathbb{P}(q_L = q_{1L} \mid o_0 \in \{0,1\}, \pi)\,\mathbb{P}(q_{L+1} = q_{+,L+1} \mid q_L = q_{1L}, \pi)) \\
&+ \frac{H - L}{2}\,\mathbb{P}(q_{L+1} = q_{+,L+1} \mid q_L = q_{bL}, \pi)
\end{aligned} \tag{51}$$

$$\begin{aligned}
= {}&\frac{H - L}{4}\,(\mathbb{P}(q_{L+1} = q_{+,L+1} \mid q_L = q_{0L}, \pi) + \mathbb{P}(q_{L+1} = q_{+,L+1} \mid q_L = q_{1L}, \pi)) \\
&+ \frac{H - L}{2}\,\mathbb{P}(q_{L+1} = q_{+,L+1} \mid q_L = q_{bL}, \pi)
\end{aligned} \tag{52}$$

$$\begin{aligned}
= {}&\frac{H - L}{4}\,(\mathbb{P}(a_L = a_0' \mid q_L = q_{0L}, \pi)\,\mathbb{P}(o_L = + \mid q_L = q_{0L}, a_L = a_0') \\
&+ (1 - \mathbb{P}(a_L = a_0' \mid q_L = q_{0L}, \pi))\,\mathbb{P}(o_L = + \mid q_L = q_{0L}, a_L = a_1') \\
&+ (1 - \mathbb{P}(a_L = a_1' \mid q_L = q_{1L}, \pi))\,\mathbb{P}(o_L = + \mid q_L = q_{1L}, a_L = a_0') \\
&+ \mathbb{P}(a_L = a_1' \mid q_L = q_{1L}, \pi)\,\mathbb{P}(o_L = + \mid q_L = q_{1L}, a_L = a_1')) \\
&+ \frac{H - L}{2}\,(\mathbb{P}(a_L = a_0' \mid q_L = q_{bL}, \pi)\,\mathbb{P}(o_L = + \mid q_L = q_{bL}, a_L = a_0')
\end{aligned}$$

$$+ \mathbb{P}(a_L = a'_1 \mid q_L = q_{bL}, \pi) \, \mathbb{P}(o_L = + \mid q_L = q_{bL}, a_L = a'_1)) \tag{53}$$

where in Eq. (52) we have used the uniform probability over $x$. Now, for any history-dependent deterministic policy $\pi$ in episodic RDPs, it is possible to identify an associated regular stochastic policy $\pi_r : \mathcal{Q}' \to \Delta(\mathcal{A})$, where $\mathcal{Q}' := \mathcal{Q} \setminus \{q_{o_\perp}\}$ and:

$$\pi_r(q, a) := \mathbb{P}(\pi(h) = a \mid \bar\tau(h) = q) \tag{54}$$

$$= \sum_{h' \in \bar\tau^{-1}(q)} \mathbb{I}(\pi(h') = a) \frac{\mathbb{P}(h = h' \mid \pi)}{\mathbb{P}(q \mid \pi)} \tag{55}$$

In other words, $\pi_r$ encodes the probability that $\pi$ takes action $a$, given that some history has led to state $q$. With this convention, we resume from Eq. (53)

$$
\begin{aligned}
\mathbb{E}[V_0^\pi(h_0)] = {}& \frac{H - L}{4} \left( \pi_r(q_{0L}, a'_0) \, v_\xi(+) + (1 - \pi_r(q_{0L}, a'_0)) \, v_\xi(-) \right. \\
& \left. + (1 - \pi_r(q_{1L}, a'_1)) \, v_\xi(-) + \pi_r(q_{1L}, a'_1) \, v_\xi(+) \right) \\
& + \frac{H - L}{2} \left( \pi_r(q_{bL}, a'_0) \, u(+) + \pi_r(q_{bL}, a'_1) \left( \mathbb{I}(c = a'_1) \, v_\eta(+) + \mathbb{I}(c = a'_0) \, v_\eta(-) \right) \right)
\end{aligned} \tag{56}
$$

$$
\begin{aligned}
= {}& \frac{H - L}{8} \left( \pi_r(q_{0L}, a'_0) \, (1 + \xi) + (1 - \pi_r(q_{0L}, a'_0)) \, (1 - \xi) \right. \\
& \left. + (1 - \pi_r(q_{1L}, a'_1)) \, (1 - \xi) + \pi_r(q_{1L}, a'_1) \, (1 + \xi) \right) \\
& + \frac{H - L}{4} \left( \pi_r(q_{bL}, a'_0) + \pi_r(q_{bL}, a'_1) \left( \mathbb{I}(c = a'_1) \, (1 + \eta) + \mathbb{I}(c = a'_0) \, (1 - \eta) \right) \right)
\end{aligned} \tag{57}
$$

$$
\begin{aligned}
= {}& \frac{H - L}{4} \left( 1 - \xi + \xi \, \pi_r(q_{0L}, a'_0) + \xi \, \pi_r(q_{1L}, a'_1) \right. \\
& \left. + \pi_r(q_{bL}, a'_0) + \pi_r(q_{bL}, a'_1) \left( 1 + \eta \, \mathbb{I}(c = a'_1) - \eta \, \mathbb{I}(c = a'_0) \right) \right)
\end{aligned} \tag{58}
$$

For the optimal policy in particular, this becomes:

$$V^* = \frac{H - L}{4} \left( 1 + \xi + \mathbb{I}(c = a'_0) + (1 + \eta) \, \mathbb{I}(c = a'_1) \right) \tag{59}$$

From the $\varepsilon$-optimality of $\pi = \mathfrak{A}(\mathcal{D})$, then,

$$\varepsilon \geq \mathbb{E}[V_0^*(h_0) - V_0^\pi(h_0)] \tag{60}$$

$$
\begin{aligned}
= {}& \frac{H - L}{4} \left( 2\xi - \xi \, \pi_r(q_{0L}, a'_0) - \xi \, \pi_r(q_{1L}, a'_1) \right. \\
& \left. + \eta \, \mathbb{I}(c = a'_1) \, (1 - \pi_r(q_{bL}, a'_1)) + \eta \, \mathbb{I}(c = a'_0) \, \pi_r(q_{bL}, a'_1) \right)
\end{aligned} \tag{61}
$$

$$= \frac{H - L}{4} \left( \xi \, (2 - \pi_r(q_{0L}, a'_0) - \pi_r(q_{1L}, a'_1)) + \eta \, (1 - \pi_r(q_{bL}, c)) \right) \tag{62}$$

$$\geq \frac{H - L}{4} \max\{ \xi \, (1 - \pi_r(q_{0L}, a'_0)), \xi \, (1 - \pi_r(q_{1L}, a'_1)), \eta \, (1 - \pi_r(q_{bL}, c)) \} \tag{63}$$

Now, assume that

$$\min\{\xi, \eta\} \geq \frac{16 \, \varepsilon}{H - L} \tag{64}$$

Then, all of the following is true: $\pi_r(q_{0L}, a'_0) \geq 3/4$, $\pi_r(q_{1L}, a'_1) \geq 3/4$, $\pi_r(q_{bL}, c) \geq 3/4$. This means that, for small $\varepsilon$, any $\varepsilon$-optimal policy must frequently select the optimal action for both the parity problem and the bandit. Let us represent the first two events with $B_p$ and the third with $B_b$. Since $\mathfrak{A}$ is $(\varepsilon, \delta)$-PAC for $\mathbb{R}(L, H, \xi, \eta)$ under $\varphi$, the probability of $B_p \wedge B_b$ is at least $1 - \delta$, for any $\mathcal{D}$ and $\pi^b$ satisfying $\varphi(\mathcal{D}, \pi^b)$.

We proceed to compute the necessary data to satisfy both events with high probability. The dataset $\mathcal{D}$ can be partitioned in two subsets $\mathcal{D}_p$ and $\mathcal{D}_b$, containing any episode from $\mathcal{D}$ whose initial observation is $\{0, 1\}$ and +, respectively. The two datasets share no information and $\mathcal{D}_p$ and $\mathcal{D}_b$ are mutually independent. To see this, we observe that the sequence $a_{L+1} r_{L+1} o_{L+1} \ldots o_H$ is independent of

$a_0 r_0 o_0 \ldots a_L$ given $o_L$, since $+$ or $-$ determines at step $L$ determines the rest of the episode. Also, for any two episodes $e_H, e'_H$, the sequence $a_1 r_1 o_1 \ldots o_L$ is independent of $a'_1 r'_1 o'_1 \ldots o'_L$ given $o_0$. Since, $o_0 \sim s$, that is the starting distribution, the two datasets are independent. Let $\mathcal{Q}_{\mathsf{p}} = \{q_s, q_{o_\perp}\} \cup \{q_{0i}, q_{1i}\}_{i=1,\ldots,L} \cup \{q_{+,i}, q_{-,i}\}_{i=L+1,\ldots,H}$ and $\mathcal{Q}_{\mathsf{b}} = \{q_s, q_{o_\perp}\} \cup \{q_{bi}\}_{i=1,\ldots,L} \cup \{q_{+,i}, q_{-,i}\}_{i=L+1,\ldots,H}$ be the reachable states in the two datasets. Then, we consider two separate classes $\mathbb{R}(L, H, \xi)$ and $\mathbb{R}(L, H, \eta)$ as the sets of RDPs in $\mathbb{R}(L, H, \xi, \eta)$, restricted to $\mathcal{Q}_{\mathsf{p}}$ and $\mathcal{Q}_{\mathsf{b}}$, respectively. To do so we construct $\mathbf{R}_r \in \mathbb{R}(L, H, \xi)$ and $\mathbf{R}_c \in \mathbb{R}(L, H, \eta)$ such that the initial observation follows $\mathrm{unif}(\{0, 1\})$ in $\mathbf{R}_r$ and $\mathbb{1}_+$ in $\mathbf{R}_c$. Now, from the independence of the two datasets and the fact that $\mathfrak{A}$ is $(\varepsilon, \delta)$-PAC in $\mathcal{D}$, there must exists an algorithm $\mathfrak{A}_{\mathsf{p}} : \mathcal{D}_{\mathsf{p}} \mapsto \pi_{\mathsf{p}}$ that is $(2\varepsilon, \delta)$-PAC in $\mathbb{R}(L, H, \xi)$ under some $\varphi_{\mathsf{p}}$, and $\mathfrak{A}_{\mathsf{b}} : \mathcal{D}_{\mathsf{b}} \mapsto \pi_{\mathsf{b}}$ that is $(2\varepsilon, \delta)$-PAC in $\mathbb{R}(L, H, \eta)$ under some $\varphi_{\mathsf{b}}$. If this was not the case, $B_{\mathsf{p}} \wedge B_{\mathsf{b}}$ could not be verified in one of the two terms.

We analyze $\mathfrak{A}_{\mathsf{p}}$ first and we show that its requirement $\varphi_{\mathsf{p}}$ is $|\mathcal{D}_{\mathsf{p}}| \in \Omega(L/\xi) \cup 2^{\Omega(L)}$. For a contradiction, assume this is not the case and that $|\mathcal{D}_{\mathsf{p}}| = g(L, \xi) \notin \Omega(L/\xi) \cup 2^{\Omega(L)}$ is allowed. Then, we can use $\mathfrak{A}_{\mathsf{p}}$ to solve the noisy parity problem under the streaming setting with $g(L, \xi)$ samples (this setting has been introduced in Appendix E.1). We proceed as follows. Consider any noisy parity function $f_{x,\xi}$ with unknown $x$. Sample a sequence of strings $\{y_i\}_i \in 2^L$ from the uniform distribution and collect $g(L, \xi)$ pairs $(y_i, p_i)$, sampling $p_i \sim f_{x,\xi}(y_i)$. Then, for $H > L$, compose a dataset of episodes $\{e_i\}_i$. All actions of $e_i$ are selected uniformly in $\{a'_0, a'_1\}$. The observations $o_{0:L-1}$ are $y_i$ and $o_L$ equals $p_i$ if $a_L = a'_0$, $1 - p_i$, otherwise (0 and 1 take roles of $+$ and $-$ symbols here). Rewards $r_{L+1:H}$ are equal to one if $o_L = 1$, null otherwise. We obtain that dataset so constructed is equally likely under this procedure than under the uniform policy and the RDP $\mathbf{R}_x \in \mathbb{R}(L, H, \xi)$. Since $\mathfrak{A}_{\mathsf{p}}$ is $(2\varepsilon, \delta)$-PAC for $\mathbb{R}(L, H, \xi)$, with probability $1 - \delta$, the output policy $\pi_{\mathsf{p}}$ satisfies:

$$\min\{\pi_{\mathsf{pr}}(q_{0L}, a'_0), \pi_{\mathsf{pr}}(q_{1L}, a'_1)\} \geq 3/4 \tag{65}$$

where $\pi_{\mathsf{pr}}$ is the stochastic regular policy for $\pi_{\mathsf{p}}$. This can be seen by our assumption in Eq. (64) and doubling both $\varepsilon$ and the sub-optimality gap of Eq. (63), due to the updated probability for the initial observation. Then, for any sequence $y \in 2^L$ and associated history $h_{L-1}$ with $o_{0:L-1} = y$,

$$f_x(y) = \arg\max_{i=0,1} \pi_{\mathsf{p}}(h_{L-1}, a'_i) \tag{66}$$

which is the noiseless parity function based on $x$. This means that it is possible to reconstruct $x$ solely by interacting with $\pi_{\mathsf{p}}$, without collecting further samples. The solution we have described is a streaming algorithm with sample complexity $g(L, \xi)$. Since this contradicts Lemma 18, we have proven $|\mathcal{D}_{\mathsf{p}}| \in \Omega(L/\xi) \cup 2^{\Omega(L)}$.

We now consider the bandit problem, which is solved by $\mathfrak{A}_{\mathsf{b}}$. Similarly to the previous case, from the $\varepsilon$-optimality of $\mathfrak{A}_{\mathsf{b}}(\mathcal{D}_{\mathsf{b}})$, we obtain the necessary condition: $\pi_{\mathsf{br}}(q_{bL}, c) \geq 3/4$ from Eq. (63). This condition is expressed for the stochastic policy $\pi_{\mathsf{br}}$, However, we notice that for $q_{bL}$ in particular, the only possible history is $h_{L-1} = a_0 + a_1 \ldots +$, where all actions must also be deterministic. Then,

$$\pi_{\mathsf{br}}(q_{bL}, c) = \mathbb{P}(\pi_{\mathsf{b}}(h) = c \mid \bar{\tau}(h) = q_{bL}) = \mathbb{I}(\pi_{\mathsf{b}}(h_{L-1}) = c) \tag{67}$$

implying that $\pi_{\mathsf{br}}$ can only be deterministic for $q_{bL}$. This means that $\mathfrak{A}_{\mathsf{b}}$ must solve best-arm identification in the two arm bandit at $q_{bL}$. We can compose a simplified dataset that is relevant for the bandit as:

$$\mathcal{D}'_{\mathsf{b}} = \{a_L o_L : e_H \in \mathcal{D}_{\mathsf{b}}\} \tag{68}$$

Since $\mathcal{D}_{\mathsf{b}}$ can be deterministically reconstructed from $\mathcal{D}'_{\mathsf{b}}$, we have the following conditional independence: $\pi_{\mathsf{b}} \perp c \mid \mathcal{D}'_{\mathsf{b}}$, where $c \in \{a'_0, a'_1\}$ is the optimal arm, and $\pi_{\mathsf{b}} = \mathfrak{A}_{\mathsf{b}}(\mathcal{D}_{\mathsf{b}})$ is the output of the algorithm. Denoting with $\hat{c} = \pi_{\mathsf{b}}(h_{L-1})$ the selected arm, the error probability is $P_e := \mathbb{P}(\hat{c} \neq c)$. Applying Fano's inequality from Theorem 12 to the variables $c \to \mathcal{D}'_{\mathsf{b}} \to \hat{c}$ gives:

$$H_2(P_e) \geq H(c \mid \mathcal{D}'_{\mathsf{b}}) \tag{69}$$

$$= H(c) - I(c; \mathcal{D}'_{\mathsf{b}}) = \log 2 - I(c; \mathcal{D}'_{\mathsf{b}}) \tag{70}$$

where we have used the fact that $\hat{c}$ is a Bernoulli variable and the uniform prior over $c$. Now, assuming $C \geq 2$, we construct the following behavior policy: $\pi^{\mathsf{b}}(q_{bL}, a_0) = 1 - 1/C$ and $\pi^{\mathsf{b}}(q_{bL}, a_1) = 1/C$. In the following, we write $N_{\mathsf{b}} := |\mathcal{D}_{\mathsf{b}}|$ and omit the implicit dependency on $\pi^{\mathsf{b}}$.

$$I(c; \mathcal{D}'_{\mathsf{b}}) = H(\mathcal{D}'_{\mathsf{b}}) - H(\mathcal{D}'_{\mathsf{b}} \mid c) \tag{71}$$

$$= N_{\mathsf{b}}(H(a_L o_L) - H(a_L o_L \mid c)) \tag{72}$$

$$= N_{\mathsf{b}} D_{\mathrm{KL}}(\mathbb{P}(a_L o_L, c) \parallel \mathbb{P}(a_L o_L)\,\mathbb{P}(c)) \tag{73}$$

$$= \frac{N_{\mathsf{b}}}{2} \sum_{a,c' \in \mathcal{A}, o \in \mathcal{O}} \mathbb{P}(a, o \mid c') \log \frac{\mathbb{P}(a, o \mid c')}{\mathbb{P}(a, o)} \tag{74}$$

$$= \frac{N_{\mathsf{b}}}{2} \sum_{a,c' \in \mathcal{A}, o \in \mathcal{O}} \mathbb{P}(a, o \mid c') \log \frac{\mathbb{P}(a \mid c')\,\mathbb{P}(o \mid c', a)}{\sum_{c''} \mathbb{P}(a \mid c'')\,\mathbb{P}(o \mid c'', a)/2} \tag{75}$$

$$= \frac{N_{\mathsf{b}}}{2} \sum_{a,c' \in \mathcal{A}, o \in \mathcal{O}} \mathbb{P}(a, o \mid c') \log \frac{2\mathbb{P}(o \mid c', a)}{\sum_{c''} \mathbb{P}(o \mid c'', a)} \tag{76}$$

$$= \frac{N_{\mathsf{b}}}{2} \sum_{a,c' \in \mathcal{A}, o \in \mathcal{O}} \mathbb{P}(a, o \mid c') \log(2\mathbb{P}(o \mid c', a)) \tag{77}$$

$$= \frac{N_{\mathsf{b}}}{2} \sum_{c' \in \mathcal{A}, o \in \mathcal{O}} \mathbb{P}(a_1', o \mid c') \log(2\mathbb{P}(o \mid c', a_1')) \tag{78}$$

$$= \frac{N_{\mathsf{b}}}{2} \sum_{o \in \mathcal{O}} (\mathbb{P}(a_1', o \mid c = a_0') \log(2\mathbb{P}(o \mid c = a_0', a_1')) + \mathbb{P}(a_1', o \mid c = a_1') \log(2\mathbb{P}(o \mid c = a_1', a_1'))) \tag{79}$$

$$= N_{\mathsf{b}}(\mathbb{P}(a_1', + \mid c = a_0') \log(2\mathbb{P}(+ \mid c = a_0', a_1')) + \mathbb{P}(a_1', + \mid c = a_1') \log(2\mathbb{P}(+ \mid c = a_1', a_1'))) \tag{80}$$

$$= N_{\mathsf{b}}\Big(\frac{1-\eta}{2C} \log(1-\eta) + \frac{1+\eta}{2C} \log(1+\eta)\Big) \tag{81}$$

$$= \frac{N_{\mathsf{b}}}{C} D_{\mathrm{KL}}(v_\eta \parallel v_0) \tag{82}$$

$$\leq \frac{N_{\mathsf{b}}\,\eta^2}{C} \tag{83}$$

Then from Eq. (70), and the fact that $\mathfrak{A}_{\mathsf{b}}$ is $(2\varepsilon, \delta)$-PAC,

$$H(\delta) \geq H_2(P_e) \geq \log 2 - \frac{N_{\mathsf{b}}\,\eta^2}{C} \tag{84}$$

$$\Rightarrow N_{\mathsf{b}} \geq \frac{C}{\eta^2}(\log 2 - H(\delta)) \tag{85}$$

Which means that this must be $\varphi_{\mathsf{b}}$, the requirement for $\mathfrak{A}_{\mathsf{b}}$.

Finally, to compose the results from both branches, we observe that $|\mathcal{D}| = |\mathcal{D}_{\mathsf{p}}| + |\mathcal{D}_{\mathsf{b}}|$. Also, for any $\delta \in (0, 0.5)$, say $1/4$, $(\log 2 - H(\delta))$ becomes a positive constant, and we can add both sizes asymptotically:

$$|\mathcal{D}| \in \Omega\Big(\frac{H}{\xi} + \frac{C}{\eta^2}\Big) \tag{86}$$

To relate the parameters to features of the RDP, we observe that the number of states of any RDP in $\mathbb{R}(L, H, \xi, \eta)$ is $Q \leq 3H$. Also, the behavior policy is uniform everywhere except in $q_{bL}$. Assuming $C \geq 2$, the computation of the single-policy concentrability coefficient yields $C_{\mathbf{R}}^* = C$, for any $c \in \{a_0', a_1'\}$. Next, we compute the $L_1^p$-distinguishability of any RDP in this class. The $L_1^p$-distinguishability of a set of states $\mathcal{Q}$ is the minimum $L_1$ distance in distribution between episodes prefixes that are generated starting from any two states in $\mathcal{Q}$. Let us consider the $L_1$ norm for the pair $q_{01}$ and $q_{11}$,

$$\|\mathbb{P}(e_{1:H} \mid q_{01}, \pi^{\mathsf{b}}) - \mathbb{P}(e_{1:H} \mid q_{01}, \pi^{\mathsf{b}})\|_1 = \tag{87}$$

$$= \sum_{e \in \mathcal{E}_{H-1}} |\mathbb{P}(e_{1:H} = e \mid q_{01}) - \mathbb{P}(e_{1:H} = e \mid q_{11})| \tag{88}$$

$$= \sum_{earo \in \mathcal{E}_L} \mathbb{P}(e_{1:L-1} = e)|\mathbb{P}(a_L = a, r_L = r, o_L = o \mid q_{01}, e) - \mathbb{P}(a_L = a, r_L = r, o_L = o \mid q_{11}, e)| \tag{89}$$

$$= \sum_{ao \in \mathcal{AO}} |\mathbb{P}(a_L = a, o_L = o \mid q_{0L}) - \mathbb{P}(a_L = a, o_L = o \mid q_{1L})| \tag{90}$$

$$= (1/2) \sum_{o \in \mathcal{O}} |\mathbb{P}(o_L = o \mid a_L = a'_0, q_{0L}) - \mathbb{P}(o_L = o \mid a_L = a'_0, q_{1L})| \tag{91}$$

$$= \quad + (1/2) \sum_{o \in \mathcal{O}} |\mathbb{P}(o_L = o \mid a_L = a'_1, q_{0L}) - \mathbb{P}(o_L = o \mid a_L = a'_1, q_{1L})| \tag{92}$$

$$= \sum_{o \in \mathcal{O}} |\mathbb{P}(o_L = o \mid a_L = a'_0, q_{0L}) - \mathbb{P}(o_L = o \mid a_L = a'_0, q_{1L})| \tag{93}$$

$$= 2|\mathbb{P}(o_L = + \mid a_L = a'_0, q_{0L}) - \mathbb{P}(o_L = + \mid a_L = a'_0, q_{1L})| \tag{94}$$

$$= 2\xi \tag{95}$$

The $L_1$ distance of suffixes from $q_{01}, q_{11}$ that are shorter than $L$ have also a distance of $2\xi$. On the other hand, any shorter prefix has a distance of 0. Since, $L_1^{\mathsf{p}}$ sums all these distances, the minimum across the $q_{0i}, q_{1i}$ pairs, is attained for $q_{0L}$ and $q_{1L}$, which determines $\mu_0 \geq 2\xi$. Also, the distance between any other pair of states in the same layer is strictly higher, since they differ deterministically in some reward or observation. Hence, the $L_1^{\mathsf{p}}$-distinguishability of the entire RDP is $\mu_0 = 2\xi$. Now, we choose $L = H/2$, $\eta = 32\,\varepsilon/H$ and we assume $\varepsilon \leq H\mu_0/64$, $H \geq 2$. We can verify that these choices are consistent with the previous assumption $\min\{\xi, \eta\} \geq \frac{16\,\varepsilon}{H-L}$. Substituting, the final requirement $\varphi$ for the complete algorithm $\mathfrak{A}$ is an exponential number of episodes in $H$ or:

$$|\mathcal{D}| \in \Omega\left(\frac{H}{\mu_0} + \frac{C_{\mathbf{R}}^* H^2}{\varepsilon^2}\right) \tag{96}$$

Now, for any $H, \mu_0, C_{\mathbf{R}}^*, \varepsilon$ satisfying the previous assumptions, any algorithm cannot be $(\varepsilon, 1/4)$-optimal for the instances in $\mathbb{R}(H/2, H, \mu_0, 32\,\varepsilon/H)$ if Eq. (96) is not satisfied. $\quad\square$

Note that in our RDP instance, the number of states and the horizon length scale linearly. So, we might equivalently write $HO$ instead of $H^2$.

