# OpenReview forum: "Provably Efficient Offline Reinforcement Learning in Regular Decision Processes"
_NeurIPS.cc/2023/Conference — NeurIPS 2023 poster_

### Official Review · Reviewer_9RiT · 2023-07-07

**Soundness:** 3 good
**Presentation:** 3 good
**Contribution:** 3 good
**Rating:** 8
**Confidence:** 3

**Summary:**

The paper presents an offline RL algorithm to learn near-optimal policies in a (episodic) Regular Decision Process (RDP). The problem is to learn a policy given a data set. The algorithm is split into two parts. First, is to learn the transition function of the RDP. Then, the problem of off-line learning on RDP is reduced to offline learning in MDP (intuitively, the underlying automata in the RDP is extracted) to generate a near-optimal policy in the RDP. The upper bound provided in this setting improves upon an upper bound of a similar algorithm in the non-episodic variant and a lower bound is also provided.

**Strengths:**

Good problem. Well-written paper. Contributions and algorithms are well-stated and well-contextualized w.r.t. related work.

**Weaknesses:**

NA.

**Questions:**

What may be some challenges in adopting these algorithms in practice?

**Limitations:**

NA.

---

> ### Author Rebuttal · Authors · 2023-08-09
>
> Thank you very much for your careful reading and recognition of our novelty. Please find our response to your questions and concerns below.
>
> ### Response to Questions:
> Input prior knowledge and assumptions are, generally, the main limiting factors to the practicality of learning algorithms. Our algorithm works under a few assumptions, as listed in the paper, and only a few inputs. Besides the input dataset, we can see that ADACT-H only requires a confidence parameter $\delta$ and an accuracy parameter $\varepsilon$, both of which are user-specified. This is a minimal set of input knowledge. The second variant, ADACT-H-A, has a different trade-off between input knowledge and assumptions. Depending on the specific application, either solution might be more appropriate than the other.
>
> We currently work under the assumption that rewards distributions are supported on a finite set. More general reward distributions are an important extension that would allow RDP to generalise to new classes of environments. We leave such a modification for future work.
>
> We will include part of this discussion in the final revision of the manuscript.

---

### Official Review · Reviewer_xyrg · 2023-07-07

**Soundness:** 3 good
**Presentation:** 2 fair
**Contribution:** 3 good
**Rating:** 6
**Confidence:** 3

**Summary:**

The authors presents a novel algorithm for offline RL in episodic Regular Decision Processes (RDP), a computationally feasible subclass of Non-Markovian Decision Processes. The presented algorithm have two phases: 1) learning the underlying automata by a novel algorithm, and 2) Markov transformation of the rest of the dataset and use any state-of-the-art algorithm for offline RL in usual MDPs. To obtain a theoretical guarantees, authors presents a notion of concentrability coefficient in the RDP setup. Finally, the lower bounds on sample complexity were presented.

**Strengths:**

- The first algorithm for offline RL in a challenging setup of Regular Decision Processes with provable guarantees.
- A novel algorithm for automata learning that have a much tighter sample complexity than in the previous work.
- The lower bounds shows that the introduced concentrability coefficient is indeed makes sense from the point of view of learning in RDB.

**Weaknesses:**

- Lack of empirical validation. The empirical experiments even on toy examples should much increase the value of the presented paper.

**Questions:**

- What are the limitation to generalize the presented techniques beyond the tabular setting?
- Is it possible to propose direct algorithm that will not depend on reduction to usual MDPs?

**Limitations:**

This is a theoretical paper that does not need to address the potential societal impact.

---

> ### Author Rebuttal · Authors · 2023-08-09
>
> Thank you very much for your careful reading and constructive suggestions. Please find our response to your questions and concerns below:
>
> ### Response to Questions:
> - We believe extension to discrete structured RDPs (with a known structure) could pose some mild challenges, in terms of algorithm design and analysis thereof, and may require some assumptions to be imposed. However, the task is certainly far from straightforward and the said challenges depend on the type of structure. In contrast, in the case of RDPs with infinite state and observation spaces, one must resort to using substantially different ideas, and we believe the problem becomes an interesting open question for future research. In the latter case, the first and most natural generalisation is considering reward distributions with continuous support. To do so, depending on the reward distribution class that is assumed, the "TestDistinct" function would need to be updated accordingly, as well as its analysis. The statistical test would be specific to continuous distribution and not be based on counting schemes.
> Although a similar modification might be done for observations, these pose an additional challenge with respect to rewards. In fact, observations are both outputs and inputs for our model. An infinite observation space would immediately imply an infinite number of input symbols and infinite potential transitions from each RDP state. This is a much more meaningful change that possibly alters the class and the expressiveness of the underlying transducer. This case would require an independent study to understand its impact.
>
> - We believe deriving a direct algorithm is a very interesting future direction that renders very challenging and calls for novel algorithmic ideas. Intuitively, such a direct algorithm avoids a sharp two-phase separation and would allow for interleaving automata learning steps with value estimations steps. This is desirable in order to direct RDP learning steps toward regions with high value estimates, thus resulting in a more sample-efficient algorithm for RDPs. However, deriving theoretical sample complexity bounds for such an algorithm could be very challenging. On the other hand, the approach taken appears natural in view of the fact that each RDP has an associated MDP, which is very easy to construct or simulate. Our current understanding is that the reduction is not a source of complexity in itself; it  was powerful enough to yield a sample complexity bound that matches some factors in the lower bound. Overall, we believe this is a very promising future direction, despite its involved challenges. Thus, we will include this discussion in the conclusion.

---

> > ### Comment · Reviewer_xyrg · 2023-08-13
> >
> > I would like to thank the authors for their answers and additional discussions. The comments have addressed my questions and I decide to keep my score.

---

### Official Review · Reviewer_9NCg · 2023-07-09

**Soundness:** 3 good
**Presentation:** 4 excellent
**Contribution:** 3 good
**Rating:** 6
**Confidence:** 3

**Summary:**

The paper presents RegORL, an algorithm for offline RL in episodic RDPs. The algorithm combines automata learning techniques with state-of-the-art offline RL algorithms for MDPs. The authors provide a non-asymptotic high-probability sample complexity bound for RegORL, which guarantees the learning of an $\epsilon$-optimal policy. They also establish a sample complexity lower bound for offline RL in RDPs.

**Strengths:**

1. The paper comprehensively and rigorously explores algorithms for offline RDPs while leveraging automata learning techniques. This contribution carries great significance within the realm of offline RL theory.

2. The structure of this paper is well-organized, resulting in a smooth and coherent reading experience. The comprehensive summary of relevant literature further enhances its overall completeness.

**Weaknesses:**

1. It would be beneficial to present a comparative analysis of RDP algorithms in tabular form, facilitating a clearer understanding of the different studies in this area.

2.  In what ways does the offline setting present additional difficulties beyond the combination of commonly used techniques from offline RL and the technique used in online RDP research [1]?





[1] Alessandro Ronca and Giuseppe De Giacomo. Efficient PAC reinforcement learning in regular decision
418 processes. In IJCAI, pages 2026–2032, 2021.

**Questions:**

1. Does the claim made by the authors in line 98, stating that computing a near-optimal policy is challenging even for a known POMDP, imply that this difficulty does not exist in the case of RDPs?

2. Can the techniques proposed in this paper be applied to settings where the state or observation space is infinite, such as RDPs with a linear structure?







**Limitations:**

The suggestions have been claimed in "Weaknesses" and "Questions".

---

> ### Author Rebuttal · Authors · 2023-08-09
>
> Thank you very much for your careful reading and constructive suggestions. Please find our response to your questions and concerns below:
>
> ### Response to Weaknesses:
> 1. As suggested by the reviewer, we will include a more explicit comparison with the other papers that address RL in RDPs. We had identified three works [25, 27, 28]. These have been discussed more extensively in the appendix. However, we will make the comparison more explicit and also include the important differences in the main body in the final version.
> 2. Offline RL in RDPs has some specific features with respect to the two, taken in isolation. One important difference is the need for a new single-policy RDP concentrability coefficient, which we defined. Its main difference with respect to the single-policy MDP concentrability is that each occupancy measure is computed on features that are not present in the dataset. Also, with regards to [1] (Ronca and De Giacomo, 2021), we mention that it addresses online RL in RDPs in a discounted, infinite-horizon setting. Importantly, this algorithm uses the uniform policy for learning. Hence, using the machinery in [1] could be adapted to our setting, only under the very strong assumption that the behaviour policy is uniform. Even in this case, such adaptation would imply sample efficiency bounds that are much looser than the ones we show here. In particular, the bound would roughly scale according to $O^{15}$ and $\epsilon^{-10}$, which is far larger than our sample complexity bounds. We will clarify this in the final version.
>
>
> ### Response to Questions:
> 1. In line 98, we mention the computational complexity of finding the optimal policy of a known POMDP. This is PSPACE-complete. For RDPs, instead, thanks to their internal structure, optimal policies can be computed very efficiently, in polynomial time. When the RDP is known, the solution is to compute the MDP of Definition 3 and find its optimal policy.
> 2. The answer differs depending on whether we consider infinite states, observations or rewards. Regarding states, the finite number of hidden states is tightly related to the finite automaton representation on which the RDP is built. This is part of what distinguishes them from more general formalisms such as POMDPs and yields the computational advantages discussed for Question 1. Such modification would strongly alter the specificity of RDPs. An infinite number of observations, on the other hand, leads to an interesting generalisation, which would allow RDPs to capture new interesting environments. The main challenge is that observations are part of the input alphabet of the transducer. In fact, an infinite number of input symbols would immediately impact the number of potential transitions from each state. This additional power modifies the class of automata under study, which would require an independent study. These are so-called symbolic automata. See, e.g., “The Power of Symbolic Automata and Transducers” by D'Antoni and Veanes, in CAV 2017. Extending RDPs to symbolic automata is an interesting direction for future work.
> On the other hand, we believe that an infinite set of rewards is a very natural extension that is worth considering. Continuous reward functions do not conflict with the finite transducer structure or its transition function. Mainly, the "TestDistinct" function would need to be updated accordingly, as well as its analysis. This is an interesting direction for future work. We include and discuss both as future directions in the revised version.

---

### Official Review · Reviewer_jsz4 · 2023-07-13

**Soundness:** 3 good
**Presentation:** 3 good
**Contribution:** 3 good
**Rating:** 8
**Confidence:** 2

**Summary:**

This work studies the problem of offline reinforcement learning (ORL) of episodic regular decision processes (ERDP) where we wish to find an near-optimal policy for an unknown ERDP with a small dataset of trajectories (collected with a behavioral policy). The authors took a reductive approach to first find the states and transitions of the minimal automata (by a proposed algorithm ADACT-H) underlying the ERDP and then to transform the original dataset into an MDP (using the estimated automata states as MDP states) and lastly to apply any existing ORL algorithm to the resulting MDP. The theoretical results are substantiated with theorems, necessary definitions and assumptions.

**Strengths:**

1. The exposition is easy to follow and the overall logic is coherent and sound.
1. The presented results support claims of contributions and seem novel.
1. The problem studied is interesting and I think that the reductive approach in this work involving recent results from automata theory (ADACT which learns a minimal Moore machine) may inspire other RL studies.

**Weaknesses:**

1. Perhaps the result of a conscious choice of tradeoff in the presentation, it seldomly highlights the key theoretical challenges and the authors' insights in the main text of the paper. Maybe the authors could reclaim some space for this purpose from example 1? (It is not referenced in the main text.)
1. In a similar vein, I wish that there is more technical discussion on (promising) un/under-explored alternative approaches and implications for related problem settings in closing. This should help contextualize the contributions of the work for a broader audience.

**Questions:**

1. The trajectory length is fixed ($H$) the episodic setting, which implies a fixed length language (unlike general automata which can accept variable length sentences). Furthermore, the automata state space is stratified over tilmestep--making the DFAs acyclic--, I wonder if this makes the full generalization of DFAs a good fit for the episodic setting.
1. Related to above, could you comment on the relevance of your approach to non-episodic setting? The DFA modeling of non-episodic environments seems more natural.
1. Is there some interesting interaction between the learned DFA and subsequent ORL? For example, does it matter if the learned DFA is slightly larger than minimal? Is there some kind of tradeoff worth sharing or further study? What are some looseness in the reduction approach you took?
1. Assumption 1, which asks for all observation to be present in the dataset, seems very restrictive (perhaps impossible for many ERDP). Since the ORL step is over $\mathcal{Q}\times\mathcal{A}$, is it possible to relax the assumption?

---

> ### Author Rebuttal · Authors · 2023-08-09
>
> Thank you very much for your careful reading, constructive suggestions, and recognition of our novelty. Please find our response to your questions and concerns below:
>
> ### Response to Weaknesses:
> As suggested, we will give further context and background about the main challenges in the introduction and closing. The discussion will include the following observations:
>
> One of the main challenges of working with non-Markovian decision processes is to develop learning algorithms that avoid exponential dependencies in the horizon in both sample- and computational-complexity. In our algorithm, we were able to achieve this result by incrementally learning each state of the RDP (all the states in ADACT-H or a subset of states in ADACT-H-A) and carefully accounting for the contribution of learning each state to the sample complexity. This role is mainly played by the distinguishability factor, which modulates the complexity of identifying each state for a specific RDP instance. In many instances, this is what allows our sample complexity analysis to remain polynomial in the relevant factors. Finally, our algorithm has low computational complexity by design.
> As future work, we believe that valuable directions can concentrate on alternative parameterizations or improvements on the definition of distinguishability. In particular, the development of a state-merging test that accounts for the (unknown) RDP structure of the distributions being tested.
>
> ### Response to Questions:
>
> 1. The automata that define Episodic RDPs are acyclic. More precisely, the class we consider is the one composed of all and only the RDP automata that generate a stop symbol after $H$ transitions. As a consequence, any automaton in this class is acyclic. This means that our work considers the most general class of automata that is consistent with a fixed horizon. This RDP definition also remains close to the ones that are present in the literature, because a non-Episodic RDP would simply be an automaton that is not forced to output $o_\bot$ after $H$ transitions.
> The explicit layered structure mainly serves to allow for some optimizations when testing RDP states. In fact, our automata learning algorithms ADACT-H and ADACT-H-A are specialised for the episodic setting, so as to have better performance than generic algorithms, which must account for the presence of cycles.
> 2. Our work is the first one addressing offline RL in RDPs. Fixed-horizon MDPs are very well-studied and popular in the theoretical RL literature (see, for example, [12, 18]), in both online and offline settings, and they appear to have relatively gained more attention than their infinite-horizon counterparts, perhaps mostly because many practical tasks of interest are episodic in nature. Our paper contributes to this line of research, extending it to RDPs. In our work, we exploit the finiteness of the horizon in two ways: both to allow some algorithmic optimizations, as discussed above, and to take advantage of the horizon in our analysis. We do agree, however, that other interactions and optimality criteria are also interesting, such as the discounted, infinite horizon setting. This would require some independent treatment that we also identified as a possible future direction of our work.
> 3. For computing an $\epsilon$-optimal policy, it is sufficient that the Offline RL algorithm receives a dataset that respects the Markov assumptions in the new states and the original rewards. To guarantee this, it is sufficient that the function "TestDistinct" never returns a false negative. Minimality of the automaton, instead, is guaranteed by the absence of false positives. This second condition, however, is of much more minor importance than the former, as it only causes an increase in the sample complexity of the offline RL part, without any impact on correctness.
> 4. Assumption 1 is only needed for the variant that learns the complete model (i.e., the one of Theorem 6), not for the approximate version of the algorithm, the one appearing in Theorem 8.
> Furthermore, Assumption 1 effectively means that the behaviour policy must select every action with positive probability, but we can modify the statement to omit the probability of observations. This is consistent with equation (2), with the convention that 0/0 = 0.
> We will improve the assumption and the surrounding text to clarify both points.

---

### Official Review · Reviewer_TAnR · 2023-07-23

**Soundness:** 3 good
**Presentation:** 3 good
**Contribution:** 3 good
**Rating:** 6
**Confidence:** 4

**Summary:**

This paper studies the sample complexity of offline reinforcement learning (RL) in environments with non-Markov observation, i.e., partially observable Markov decision processes (POMDPs). In specific, the paper considers the episodic regular decision process (RDP) where the spaces of state, action, and observation are all finite. Given a batch of offline data, the proposed (offline RL) algorithm learns the ''states'' from the history data and converts the problem as offline RL for MDPs. The paper provides sample complexity bound for learning the states, as well as the optimal policy under certain assumptions. The paper also provides lower bound to prove the optimality of the upper bounds.

Overall, the reviewer feels that the proposed algorithm for POMDPs brings new insight to the RL community and the theoretical results are strong enough to be accepted, even though they are built on strong assumptions, for example the distinguishability condition. As claimed by the authors, this is the first RL complexity bound of the setting and approach. Please see more details in the strengths and weakness parts.

**Strengths:**

The episodic regular decision process is conceptually a very large set of models that can be used to model many realistic environments. The 'state' the proposed algorithm trying to learn is essentially a sufficient statistics of the history for inference of future observations (folloiwing the behavior policy), which is very similar to the predictive stats representation (PSR) but slightly different in key assumptions (and leading to different learning approach). If my understanding is correct, every episodic finite-state POMDP is also an episodic RDP since we can always define the concatenation of the entire history or the belief, i.e., conditional distribution, of the 'hidden state' as the 'state' of the RDP.

**Weaknesses:**

1. The methodology that we first learn the state and then apply RL algorithm to the transformed data is quite a quite a straightforward approach. And it seems that Assumption 2 is the critical reason that we can eventually obtain the polynomial sample upper bound $\sqrt{H}log(QAO)/(d\mu)$ following such a straightforward approach. My biggest concern will be the scale of $\mu_0$ which is related to the distinguishability. In particular, $\mu_0$ represents the $L_\infty$ distance between distributions over $e_{t:H}$, which will be $(OA)^{H-t}$-dimensional vectors with bounded $L_1$ norm. Roughly speaking, it feels that the average $L_\infty$ distribution will be $(OA)^{-(H-t)}$. Then, unless the maximal distance $\mu_0$ has a jump in the orders due to some special structure of the RDP, we seem to have exponential dependency on $H$ in the sample complexity. Perhaps more discussion of the scale of $\mu_0$ with examples would be necessary.
2. Assumption 1 is also a very strong assumption and greatly reduces the difficulty of learning the states and optimal policy. I am wondering if we can still obtain the current strong results by adding pessimism (e.g., as in [18]) into the current algorithm and removing Assumption 1.
3. More technical discussion about the (reason for) improvement compared with the previous results [10, 27] in the main paper would be helpful for readers to understand the contribution of the current paper better.

**Questions:**

-

**Limitations:**

-

---

> ### Author Rebuttal · Authors · 2023-08-09
>
> Thank you very much for your careful reading and constructive suggestions. Please find our response to your questions and concerns below:
>
> 1. In the worst case scenario, the reviewer's intuition is correct. The distinguishability parameter in $L_\infty$ distance may be exponentially small in the remaining horizon. This is indeed the case in the hard instance used for proving the lower bound. In the RDP of Figure 3 of the appendix, the two paths $q_{01} \dots q_{0L}$ and $q_{11} \dots q_{1L}$ are indistinguishable for $L-1$ steps, when they generate strings with uniform probabilities, and differ only after $L$ steps. Each string of length $L$ has a probability of $2^{-L}$ of being generated.
> However, this does not imply that distinguishability is necessarily small. In fact, two factors contribute to a larger parameter $\mu_0$, as desired:
>     - First, we recall that we compute distinguishability with respect to the *prefix* $L_\infty$ distance, which we denote by $L_\infty^\mathsf{p}$. The prefix distance computes the maximum distance between distributions over strings of all lengths. This means that if two states can be separated just by looking at suffixes of length $K$, where $K$ could be as small as 1, for effects that are immediately observable, the distinguishability parameter for those two states will scale as $2^{-K}$, in the worst case, not $2^{-H}$.
>     - A second factor that allows $\mu_0$ to be large is the existence of one or few witnesses for distinguishability: these are traces of any length in $[1,H]$ that have high probability of being generated from one state but not from the other. Looking at the extreme side of the spectrum, we can consider RDPs with deterministic outputs, which are still non-Markovian environments. In this case, the prefix $L_\infty$-distinguishability coincides with $\max_{u \in [t]} \sum_{e_{0:u}} \left| p_1(e_{0:u} *) - p_2(e_{0:u} *) \right|$.
> More generally, all the intermediate cases are possible.
>
>
> 2. Assumption 1 is only needed if the full algorithm is combined with ADACT-H, which learns the complete RDP. The approximate version, that is ADACT-H-A, the algorithm referred to in Theorem 8, does not require Assumption 1. We will clarify this fact in the camera-ready version.
> However, integrating the two phases into a single learning algorithm is one of the main future directions we had also identified, which we will include in the final version. This will allow us to propagate low-value estimates from the RL side into the automata learning algorithm.
> 3. We will clarify the differences with respect to the mentioned references in the final version. To briefly highlight the main observations here, we can comment for [10] and [27] separately.
>     - The paper [27] addresses online RL of RDPs in a discounted, infinite-horizon setting. Importantly, this algorithm uses the uniform policy for learning. So, the algorithm might be adapted to our setting, only under the assumption that the behaviour policy is uniform. Even in this case, such adaptation would imply sample efficiency bounds that are much looser than the ones we show here. In particular, the bound would roughly scale according to $O^{15}$ and $\epsilon^{-10}$.
>     - The paper [10] is an important reference for the automata learning algorithm that we adapted to the episodic setting. Specifically for this part, we compare the upper bound at the bottom of page 7, and our upper bound in equation (4). Improving over known bounds for automata learning is necessary in order to obtain tight bounds for RL  so as to identify the true sources of complexity and hence obtain a better understanding of the problem overall.

---

> > ### Comment · Reviewer_TAnR · 2023-08-20
> >
> > Thanks for authors' clarification to my concerns and further explanations. I will keep my score as weak accept.

---

### Decision · Program_Chairs · 2023-09-21

**Decision:**

Accept (poster)

**Comment:**

This work considers offline RL in a subclass of finite POMDPs. In this class called regular decision processes the observations can be captured by a finite state automaton. The paper provides sample complexity bound for learning the process and the optimal policy under certain assumptions. This bound is complemented with a tight lower bound.

All reviewers are in consensus that this paper tackles an interesting setting and although the approach perhaps follows expected techniques, the results are significant and well executed. The paper is also clear and well written. The AC agrees with this assessment and hence recommends acceptance.